# Demographics and recovery potential of exploited marine teleosts

David M. Keith[1]*, Heather D. Bowlby[1], Camille Albouy[2,3], Arnaud Auber[4],
Danielle M. Baribeau[5], Daniel G. Boyce[6], Julie A. Charbonneau[7], Freya Keyser[1,5],
Kristin M. Kleisner[8], Martin P. Marzloff[9], Katherine E. Mills[10], David Mouillot[11],
Aurore Receveur[11], Nancy L. Shackell[1], Rita P. Vasconcelos[12]

1 Fisheries and Oceans Canada, Bedford Institute of Oceanography, Dartmouth, Nova Scotia, Canada,
2 Ecosystems and Landscape Evolution, Department of Environmental Systems Science, ETH Zürch,
Zürich, Switzerland, 3 Swiss Federal Institute for Forest, Snow and Landscape Research WSL,
Birmensdorf, Switzerland, 4 Ifremer, HMMN, Laboratoire Ressources Halieutiques, Boulogne-sur-Mer,
France, 5 Department of Biology, Dalhousie University, Halifax, Nova Scotia, Canada, 6 Wild Ocean
Research, Cow Bay, Nova Scotia, Canada, 7 Department of Biological Sciences, Simon Fraser University,
Burnaby, BC, Canada, 8 Environmental Defense Fund, Boston, United States of America, 9 Ifremer
DYNECO, Plouzané F-29280, France, 10 Gulf of Maine Research Institute, Portland, Maine, United
States of America, 11 MARBEC, Univ Montpellier, CNRS, Ifremer, IRD, Montpellier, France, 12 IPMA,
Portuguese Institute of the Sea and Atmosphere, Portugal

* david.keith@dfo-mpo.gc.ca

pone.0340369

University of Thessaloniki, GREECE

**Peer Review History:** PLOS recognizes the
benefits of transparency in the peer review
process; therefore, we enable the publication
of all of the content of peer review and
author responses alongside final, published
articles. The editorial history of this article is
available here: https://doi.org/10.1371/journal.
pone.0340369

## Abstract

Equilibrium concepts and the expectation of compensatory density dependence
remain fundamental to fisheries science, but stock collapses and an increasing
appreciation of environmental factors have raised questions about their real-world
applicability. To explore the demographic variability of harvested marine fishes, we
have calculated metrics commonly used in conservation biology to describe the
demographics for 77 assessed stocks from the North Atlantic and Northeast Pacific
Oceans using life-tables. We found that median annual population growth rates ($\lambda$
) were centered around 1, and surprisingly, they were only slightly higher when the
effect of fishing was excluded. For most stocks, as abundance declined, $\lambda$ tended
to increase and become more variable as would be expected from compensatory
dynamics. The population growth of several stocks was sustained by a limited num-
ber of years with exceptionally high rates. However, the ability of a stock to increase
from low abundance appeared largely independent of life history characteristics and
exhibited stronger geographical differences among stocks of the same species (nota-
bly Atlantic cod). Life history characteristics alone were poor predictors of annual
population growth or future recovery potential, whereas regional factors appeared to
be more influential. Overall, recovery potential remained relatively high, with simu-
lations indicating that 62 of the stocks would be highly likely to double in size within
20 years in the absence of fishing. Low recovery potential was exclusively observed
in stocks with a low median $\lambda$ and low variability in $\lambda$. These results suggest that
understanding stock-specific (rather than species-specific) demographic parameters

**Data availability statement:** Data is available in the following GitHub repository https://github.com/JulieCharbonneau/Age-structured-marine-fish-database.

**Funding:** The author(s) received no specific funding for this work.

**Competing interests:** The authors have declared that no competing interests exist.

is necessary to promote sustainable management or develop rebuilding plans for collapsed stocks.

## Introduction

Marine fishes were once believed to be inexhaustible and immune to the effects of fishing [1–2]. However, striking collapses of once-prolific fisheries and their delayed or failed recovery [3–5] have contradicted this perception. The assumption of resilience originated from early theoretical population dynamics modeling (e.g., [6]) that culminated in the development of modern fisheries science, largely built around the concept of maximum sustainable yield (MSY, [7–9]). Despite long-standing concerns regarding the use of MSY [10,11], its theoretical underpinning in equilibrium concepts and the associated ideas that annual per-capita reproductive output and/or juvenile survival rates increase as abundance declines have persisted in fisheries science [8].

Indeed, many modern assessments incorporate a stock-recruit function in which per-capita recruitment increases as biomass declines [7,9,12,13]. Such fisheries assessments implicitly assume that lifetime reproductive rates in the absence of fishing should be greater than one [12,14] and that even heavily-exploited fish stocks retain an innate capacity for recovery [15]. Although MSY and equilibrium concepts are relatively straightforward, intuitive, and provide a means of developing tractable mechanistic models, questions remain regarding their applicability [10,11]. There are concerns about whether they accurately reflect the system being modeled [12,16,17], particularly for species with markedly different life-history characteristics [18].

Life-history characteristics (e.g., longevity, maximum size, age at maturity) are often considered indicative of a species' natural mortality rate, reproductive capacity and overall productivity [19–21], and are increasingly used within data-limited assessment approaches to provide management advice [22,23]. Ecological theory suggests that 'r-selected' species characterized by small body size, numerous offspring, rapid growth, early maturity and short lifespans, should have greater recovery potential and be better able to withstand fishing pressure [24,25]. Conversely, "K-selected" species with life-histories characterized by longer lifespans and later maturity, or those with lower per-capita reproductive output, are considered less resilient to fishing pressure and are expected to have a lower recovery potential [21]. The life-history traits of 'r-selected' species should manifest as a high innate capacity for population growth in the absence of fishing, resulting in rapid recovery from low abundance [26]. Furthermore, owing to a shared evolutionary history, the life-history traits of closely related species are anticipated to be more similar than those of distantly related species [27]. This implies that there should be less heterogeneity in the productivity among stocks of a single species than between stocks of different species with similar life-history traits. However, it remains unclear if correlated and compounding ecological and environmental processes (e.g., climate change effects, predator-prey dynamics) can obscure the importance of life-history on realized productivity. This raises concerns that evolutionary history and life-history characteristics may serve as a weak foundation for estimating demographic rates of species for which this information is not available.

Stock assessment data from commercially-harvested marine fishes provide a unique opportunity to explore demographic variability and realized compensatory responses. Focusing on species with a long assessment history means that their life-history characteristics are likely to be well-known and that it is possible to quantify realized variability in survival and reproductive output while accounting for the impact of fishery removals. We based our analyses on 77 marine fish stocks included in a database of age-specific data [28]. By constructing life-tables (i.e., age-specific estimates of survival and reproduction rates) from abundance-at-age and catch-at-age data, we calculated standard demographic metrics, including annual population growth rates, doubling time, lifetime reproductive success, and generation time [29]. Our objectives are to: 1) quantify the potential for population growth ($\lambda$) and the strength of density dependence for each stock, 2) quantify recovery potential of each stock, and 3) explore the variability of these demographic rates both regionally and taxonomically. Our results have implications for the sustainable management of these commercially-harvested marine fishes and could be used to inform future management actions for stocks that have collapsed.

## Methods

### Data

These analyses were conducted using stock assessment data from a repository which included sufficient data for 77 stocks from the Northeast Atlantic, Northwest Atlantic, and the North Pacific; including stocks managed by International Council for the Exploration of the Sea (ICES; 44 stocks), NOAA (NOAA;25 stocks), and Fisheries and Oceans Canada [DFO, 8 stocks, Fig 1; this repository has been updated with the data used in this analysis [28,30]. Data were available between 1947 and 2018, with an average time series length of 38.6 years (range 13–68 years). The assessed stocks represented 6 taxonomic orders, with the majority of stocks representing Gadiformes (39 stocks), Pleuronectiformes (20 stocks) and Clupeiformes (11 stocks). The other three orders had relatively few stocks included in the analysis, with 4 Perciform stocks, 2 Scombriform stocks, and just 1 Scorpaeniform stock. The majority of the data were for stocks from six species: Atlantic cod (*Gadus morhua*; 14 stocks), herring (*Clupea harengus*; 9 stocks), haddock (*Melanogrammus*

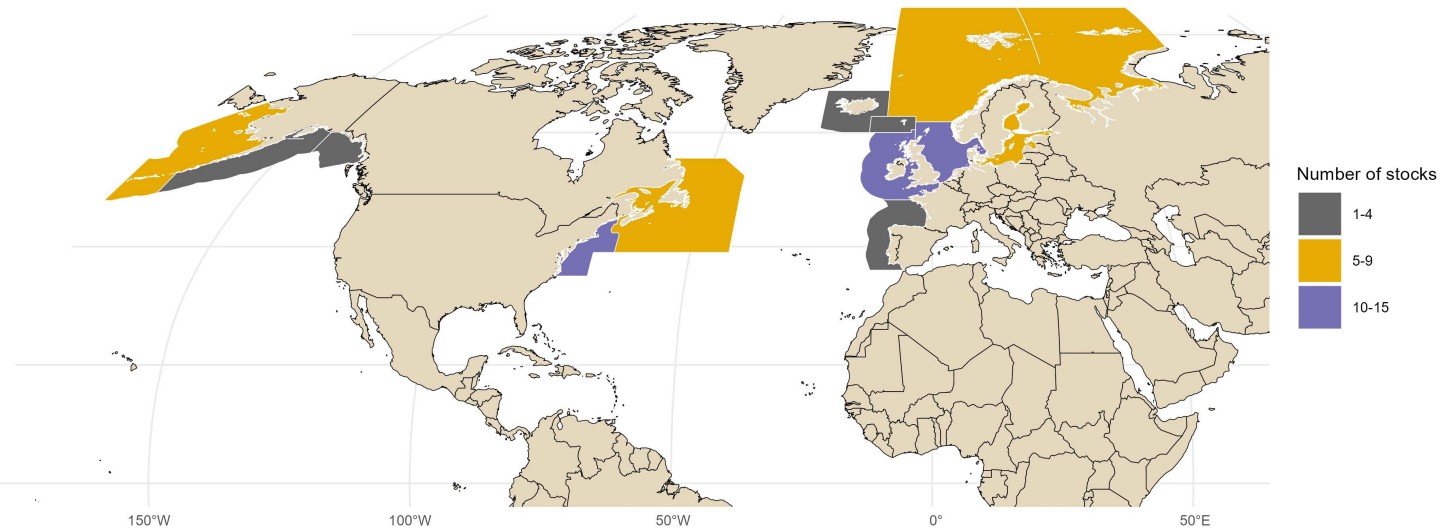

**Fig 1. Locations of the stocks used in this analysis.** Colors indicate the number of stocks within each subarea. In the Atlantic the subareas were divided using the ICES and NAFO subareas. The ICES region was grouped into seven subareas which broadly aligned with the ecoregions definitions developed by ICES. The NAFO subareas were divided into two regions, the first combining NAFO Regions 2J, 3, and 4 with the second combining NAFO regions 5 and 6(A-C). In the Pacific, the subareas were divided into a Gulf of Alaska subarea and a subarea covering the Bering Sea and Aleutian Islands. The divisions were made using the Alaska Department of Fish and Game Commercial Fisheries Groundfish Registration Areas.

*aeglefinus*; 8 stocks), pollock (*Pollachius virens*; 5 stocks), common sole (*Solea solea*; 5 stocks), and European plaice (*Pleuronectes platessa*; 4 stocks).

The repository included age-specific estimates of abundance, weight (kg), biomass (tonnes), maturity, natural mortality, and fishery removals (tonnes) for each stock (Table A1). While uncertainty is inherent in assessment processes, each assessment is the basis for management advice. As such, the underlying data represents the best available understanding of each stock's population dynamics at the time of the assessment.

## Population growth rates

We used life-table reconstructions to calculate survival-at-age and fecundity-at-age over time from the available data series. We applied the Euler-Lotka equation [31] to estimate the corresponding population growth rate ($\lambda$) in a given year. This allowed us to use information on recruitment, mortality, and maturity to describe each stock's demographics and how they changed with age and time using a tractable and consistent model.

The discrete time Euler-Lotka equation was first proposed by Euler [later published by 32] and is a special case of the continuous time demographic model subsequently developed by Lotka [31,33,34]:

$$1 = \sum_{a=\nu}^{\omega} \lambda_y^{-a} s_{y,a} f_{y,a} \Delta x$$

(1)

Where $a$ is the age, $\nu$ is the age at first reproduction, $\omega$ is the maximum age, $\lambda$ is the discrete rate of population growth ($\lambda = e^r$, where $r$ is the intrinsic rate of increase), $s_{y,a}$ is the survivorship (proportional) to age $a$, and $f_{y,a}$ is the fecundity of individuals of age $a$. This equation is solved for each year ($y$) with available data. The discrete-time Euler-Lotka equation is the characteristic polynomial of the Leslie Matrix, where $f_0$ is the fecundity of the first age class with mature individuals [29,35]. We calculated survivorship-at-age and fecundity-at-age using the available stock assessment data, which enabled $\lambda$ to be estimated. This assumes that each stock represents a discrete population with independent dynamics.

When calculating survival and fecundity inputs, we treated the last observed age class as the terminal age, beyond which there was no contribution to productivity. The first age class was age 1 for the majority of stocks (N=42), where only 21 of the stocks had a first age of 2 (N=10) or greater (N=11). For simplicity, we assumed survivorship to the first age class represented by the data to be 1, which can result in a slight overestimation of productivity from the Euler-Lotka equation [36]. However, we tested the influence of this assumption by back-calculating the abundance for the earliest ages using the available time series of age-specific natural mortality rates, and this resulted in negligible differences to the annual productivity estimates (Figs S10 and S11).

**Fecundity-at-age.** We parameterized fecundity as the number of offspring (recruits) produced by each female in each age class rather than as the number of eggs, because egg production by size, fertilization rates, and egg-to-age-0 survival rates are unknown. We first calculated an overall recruitment rate as the number of recruits produced per kilogram of spawning biomass $RPS_y$:

$$RPS_y = \frac{R_y}{SSB_{y-b}}$$

(2)

Where $R_y$ is the number of individuals in the youngest age class available for a specific stock (hereafter referred to as recruits), and $SSB_{y-b}$ is the total spawning stock biomass in year $y$, offset by the recruit age ($b$). For example, if the first age class in the abundance time series is 2 years, we assumed that the recruits in 2010 were produced by the spawning stock biomass in 2008 ($SSB_{y-b}$).

Next, we calculated the total number of recruits produced by each age class ($RPA_{y,a}$) by multiplying the spawner biomass in each age class $SSB_{(y-b),a}$ by the number of recruits produced per kilogram of spawning stock biomass:

$$RPA_{y,a} = SSB_{(y-b),a}RPS_y \tag{3}$$

The reproductive success of fish is expected to vary by age [37–39]. However, due to a lack of available evidence to quantify these effects, we made the simplifying assumption that each kilogram of spawning biomass contributed equally to the total reproductive output, regardless of age. This assumption would have a negligible impact on the overall $\lambda$ estimates, but would have an effect on the relative contribution of the underlying demographic parameters (i.e., the fecundity and mortality contributions). We calculated the age-specific fecundity ($f_{y,a}$) as the number of offspring produced by each spawner in the stock. The number of spawners in year $y$ and age class $a$ ($NS_{y,a}$) is the proportion of mature individuals in each age class ($Mat_{y,a}$) multiplied by the number of individuals in each age class:

$$NS_{y,a} = Mat_{y,a}N_{y,a} \tag{4}$$

where annual fecundity becomes:

$$f_{y,a} = \frac{RPA_{y+1,a}}{NS_{(y-b+1),a}} \tag{5}$$

Fecundity is offset by one year, as the recruits observed in year $y+1$ enter into the Euler-Lotka formulation in year $y$.

**Survivorship-to-age.** The number of individuals surviving to each age (survivorship-to-age) depends on age-specific mortality rates (i.e., survival-at-age). Commercially exploited fishes experience fishing mortality ($F$) once they grow large enough to become vulnerable to the fishing gear [40]. Fishing mortality ($F$) and natural mortality ($M$) in combination yield the total mortality ($Z$) affecting each age class in each year. Due to challenges with disentangling natural mortality and fishing mortality (known as separability) different assumptions are made about natural mortality during stock assessments. For example, natural mortality may be assigned a different value for each age (e.g., Bering Sea *Gadus chalcogrammus*), allowed to vary over time and by age (e.g., NAFO 5–6 GB *Gadus morhua*), input as a constant across ages (e.g., NAFO 5–6 *Pollachius virens*), or specified via another method. These natural mortality estimates are derived within the assessment model or estimated empirically using various methods [41]. We did not explore various ways of calculating $M$ and used the estimates of natural mortality by age and year available in the data repository (representing outputs from each assessment), which we converted into annual rates at-age for input into the Euler-Lotka equation. Both natural mortality and fishing mortality were calculated as proportional values.

The data also contained the number of fish harvested by the fishery each year and for each age class. To estimate fishing mortality ($F$) for each age class in each year we divided the catch-at-age data by the assessed numbers at-age.

$$F_{y,a} = \frac{Catch_{y,a}}{N_{y,a}} \tag{6}$$

Together, fishing mortality ($F_{y,a}$) and natural mortality ($M_{y,a}$) provide a total mortality estimate ($Z_{y,a}$) for each age in each year.

$$Z_{y,a} = F_{y,a} + M_{y,a} \tag{7}$$

We defined survival-at-age as:

$$l_{y,a} = 1 - Z_{y,a} \tag{8}$$

Next, we used survival-at-age to calculate cumulative survivorship-to-age used as an input for the Euler-Lotka equation:

$$s_{y,a} = s_{y,(a-1)} \times l_{y,(a-1)} \tag{9}$$

The availability of catch-at-age time series in the data repository allowed for the separation of fishing and natural mortality rates, enabling estimates of $F$ and $M$ (as above). As a result, we were able to calculate two $\lambda$ values, the realized $\lambda_{real}$ from the reconstruction that accounted for $F$, $M$, and fecundity, and a theoretical $\lambda$ value using the $M$ and fecundity estimates from the reconstruction. Given this second $\lambda$ value excluded the impact of fishing, and we refer to this as the demographic lambda ($\lambda_{dem}$). We recognize that the age structure in the underlying time series is still influenced by fishing pressure, and therefore $\lambda_{dem}$ does not represent the maximum intrinsic rate of growth in the absence of fishing [42]. The value is the potential growth of the population if the 'surplus production' was not harvested in a given year. Thus, $\lambda_{dem}$ represents the productivity that the stock could have achieved in each specific year, if there had not been fishing.

**Estimation.** It is not straightforward to propagate observation error in each time series when calculating survivorship-to-age and fecundity-at-age. To ensure that variability did not unduly affect predictions for annual stock productivity, we fit the Euler-Lotka model using an iterative minimization process, where predicted abundance in the following year:

$$N_{y+1} = \lambda_y N_y \tag{10}$$

was compared to the abundance estimate from the stock assessment for the corresponding year ($N_{y+1}$).

Where necessary, we adjusted the annual age-specific fecundity and natural mortality estimates until the $\lambda$ estimate from the Euler-Lotka equation resulted in an abundance time series estimate ($N_{y+1}$) that was within 5% of the stock assessment abundance time series (median difference was −0.1%). This adjustment was performed iteratively, with a step change (for most stocks each step was a change of 0.5%, but 0.05% was used for some stocks) first applied to the natural mortality estimates at each age. If the estimated abundance was not within 5% of the observed abundance estimate, then a step change in the fecundity was applied to each age. This procedure was repeated until the estimated abundance was within 5% of the observed abundance.

This approach ensured that our analyses reflected the same understanding of stock productivity, natural mortality and fecundity as the current assessments, by effectively treating the abundance estimates in the data repository as the observations. Although there are inherent limitations to treating stock assessment outputs as observed data [43], these assessments represent the best available knowledge on the realized population dynamics of each stock. Thus, abundance time series from the stock assessments are an appropriate benchmark against which to compare.

The minimization procedure resulted in the final estimates of $\lambda$ being approximately 6.7% smaller than the initial estimates. Population growth rates from life-table analyses typically decline when variability in the input parameters is accounted for (e.g., [44]). However, it is also possible that the removals time series used in the stock assessments did not capture all sources of fishing mortality (e.g., unreported catch, bycatch and incidental mortality, [45], [46]). This would result in the estimates of $\lambda_{dem}$ being lower than reality, necessitating slight reductions in fecundity and/or increases in natural mortality during the iterative minimization process.

## Density dependence

Life-table analyses make no assumptions about population regulation and thus do not incorporate a density-dependent relationship between spawners and recruits [47]. However, our estimated productivity parameters could be explored for evidence of density dependence by modelling each stock's annual population growth rate against relative stock size (expressed as a proportion of maximum abundance) in the same year. Density independence occurs when $\lambda$ shows no relationship with abundance. Negative density dependence is observed when $\lambda$ declines as abundance increases; this is the type of compensatory response described by typical spawner-recruit relationships [12]. Positive density dependence

occurs when $\lambda$ declines with declining abundance, which is also called an Allee effect or depensation when it occurs at low abundance [14,48–50]. We evaluated both the realized productivity estimates ($\lambda_{real}$) and the theoretical demographic productivity values excluding the impact of fishing mortality ($\lambda_{dem}$) for evidence of density dependence.

We used Generalized Additive Mixed Models [51,52] to describe potential relationships between $\lambda$ and abundance:

$$\lambda_{i,j} \sim N\left(0, \sigma_{i,j}^2\right)$$

$$E\left(\lambda_{i,j}\right) = \mu_{i,j}$$

$$log\left(\mu_{i,j}\right) = Prop_i + Stock_j \times Prop_i$$

$$Stocks_{i,j} \sim N\left(0, \sigma_{Stocks}^2\right) \tag{11}$$

Where $Prop_i$ is the proportion of the maximum observed abundance for the $i^{th}$ observation. The stock ($Stock_j$) is assumed to be a random effect, which is normally distributed with a mean of 0 and a variance of $\sigma_{Stocks}^2$. The response variable $\lambda$ is assumed to follow a normal distribution on the log scale. The $Stock$ by $Prop$ interaction was modeled using the $fs$ basis in the 'mgcv' package in R (version 4.5). This formulation treats the smooth curves as random effects; it is used when a factor has a large number of levels, and it enables each stock to have a unique smooth [52,53]. We explored various model formulations and alternative error distributions; however, all models had similar challenges overcoming either the underlying non-normality or the heteroskedasticity of the data. Given these challenges, we did not use the model to determine the statistical significance of the observed trends (e.g., we did not evaluate if the smooth terms are significant). This approach was adopted to capture the global relationship and provided a relative metric of variability among the stocks. We compared values of $\lambda$ estimates for each stock at the minimum, maximum, and 40% of the maximum abundance; the latter value was chosen based on both theoretical expectations and observed population dynamics [e.g., 50,51,54]

For a life-table reconstruction such as this, the maximum value of $\lambda$ is constrained at higher abundances because the maximum abundance is a known value (e.g., the maximum possible $\lambda$ at 50% of the maximum abundance is 2, at 25% it is 4, and at 100%, it is 1). Conversely, the minimum value of $\lambda$ is constrained at lower abundances (e.g., $\lambda$ must be $\geq 1$ when the stock is at its minimum observed abundance). As a result, at the maximum abundance $\lambda$ will be $\leq 1$, at the minimum $\lambda$ will be $\geq 1$, and higher variability in $\lambda$ values at low abundances should be relatively common.

### Recovery potential

Numerous demographic parameters can be derived from the outputs of the life-table reconstruction [29]. Here, we focus on three that are strongly linked to recovery potential: lifetime reproductive success, generation length, and doubling time [55].

**Lifetime reproductive success.** Lifetime reproductive success, also called the net reproductive rate, describes the average number of female spawners produced by one female spawner throughout her lifetime. Values greater than 1 suggest a stock increased, values less than 1 suggest a stock declined, and a value of 1 indicates that the average female has produced sufficient offspring over her lifetime to replace herself [29]. Lifetime reproductive success ($R_0$) was calculated for each cohort in which fecundity and natural mortality were available for complete cohorts (i.e., cohorts in which all available age classes are represented), assuming a 50:50 sex ratio.

$R_0$ was calculated both by using only the natural mortality component ($F = 0$) and with the combined effect of natural mortality and fishing mortality ($F > 0$), respectively. This provided a measure of the reproductive potential from observed stock characteristics alone, and how it changed with fishing pressure, recognizing that setting $F = 0$ does not fully remove the effect of fishing pressure on the cohort. $R_0$ was calculated for each cohort as:

$$R_0 = \sum_{x=\alpha}^{\omega} s_x f_x \tag{12}$$

This formulation ignores any contribution from a plus group (i.e., the effect of individuals who are older than the oldest age in the life-table) by assuming that the last observed age in the data represents the longevity of the cohort.

**Generation length.** Generation length is an alternate indicator for the relative reproductive rate of a stock, where shorter generation lengths are associated with faster population growth. Here, we calculated generation length for each cohort as the mean age of spawners that produced recruits [29]. Values of both the lifetime reproductive rate and generation length were derived for each complete cohort, which meant that we could also evaluate changes in reproductive capacity over time for each stock. Additionally, we compared theoretical reproductive potential and generation lengths from life history characteristics alone ($F = 0$) with those expected under current fishing mortality ($F > 0$).

We defined generation length as the mean age of the spawners that produced the observed recruits in a cohort ($L_G$, [29]).

$$L_G = \frac{\sum_{x=\alpha}^{\omega} x s_x f_x}{\sum_{x=\alpha}^{\omega} s_x f_x} \tag{13}$$

**Doubling time.** Comparing the median time frame over which abundance can double (i.e., doubling time) in the absence of fishing provides an indication of relative productivity among species and stocks. We applied a Monte Carlo approach that draws from the observed distribution of fecundity and natural mortality in a projection simulation to estimate the doubling time for each stock.

The annual fecundity and natural mortality values varied considerably across stocks in this study. Therefore, we used a simulated stock projection to account for this variability when evaluating recovery potential. Across 1000 simulations, we took a random sample of the annual values for fecundity and natural mortality, used the life-table reconstruction to estimate $\lambda$, and calculated abundance in the following year. Abundance of each stock was projected over 100 years, and the proportion of the simulations in which the stock size had doubled from the initial abundance was recorded each year. We considered stocks to have high recovery potential if 75% of simulated trajectories doubled after 10 years, medium potential if 50–74% of simulations doubled after 20 years, and low recovery potential if < 50% of simulated trajectories doubled after 20 years. For simplicity, we did not incorporate density-dependent effects or any correlation between fecundity and natural mortality. Once a population had doubled, it was considered to have remained at that abundance or greater for the remainder of the projection. This indicates that the metric reflects the earliest possible doubling time for each simulation in the absence of fishing mortality.

## Results

### Population growth rates ($\lambda$)

The median population growth rate when excluding the effects of fishing ($\lambda_{dem}$) for the 77 stocks in this analysis was 1.091, representing an innate capacity for annual population growth of approximately 8.7%. Aggregating over years, the stock with the highest median theoretical population growth rate was Atlantic cod in the Irish Sea (median $\lambda_{dem} = 1.44$) and the lowest was pollock in the Gulf of Alaska (median $\lambda_{dem} = 0.89$; Table 2). When the effects of fishing were accounted for, the overall median population growth rate ($\lambda_{real}$) dropped to 0.971, corresponding to overfishing by 2.9% annually (Figs 2 and 3).

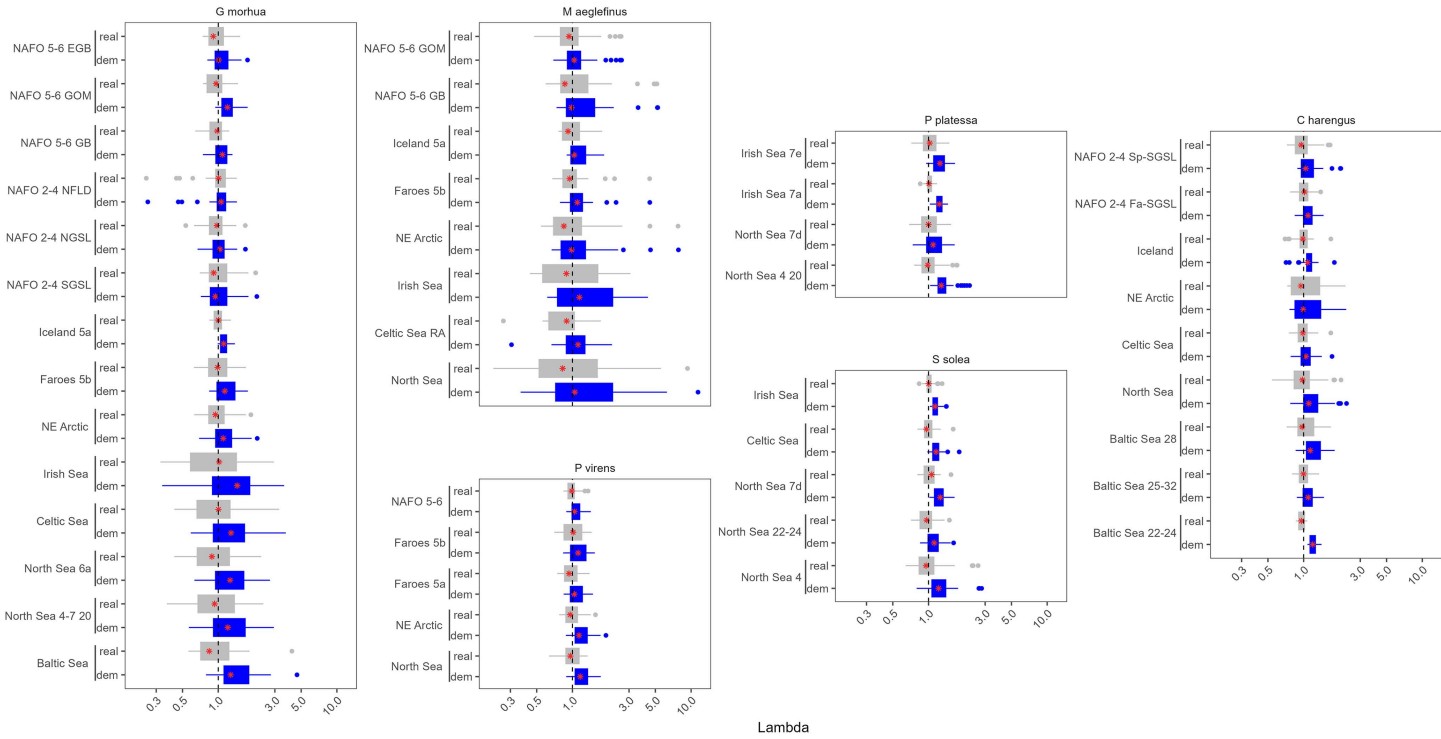

**Fig 2. Boxplot of the $\lambda$ estimates for each stock of *G. morhua*, *M. aeglefinus*, *P. virens*, *C. harengus*, *P. platessa*, and *S. solea*. Blue are the $\lambda_{dem}$ estimates and grey are the $\lambda_{real}$ estimates.** The red stars indicate the median and the dashed vertical line represents a $\lambda$ of one. The x axis is on the log scale.

The realized range of population growth rates was much narrower, and the median $\lambda_{real}$ was below 1 more than 50% of the time in 85.7% of the stocks (Figs 2 and 3). However, many of the stocks included in our analysis experienced infrequent large recruitment events that resulted in high $\lambda_{real}$ estimates (Figs 2 and 3). For example, while the Gulf of Alaska pollock stock had the lowest median $\lambda_{dem}$ (0.89), it had a mean $\lambda_{dem}$ of 1.22, and when the effects of fishing were included, this stock had the 8$^{th}$ highest mean $\lambda_{real}$ (1.19, Figs 2 and 3).

Mean $\lambda_{real}$ was lower than the median for just 8 of the 77 stocks, and the mean $\lambda_{real}$ was less than one for 15 of the 77 stocks, indicating that years of high population growth resulted in a net positive productivity for most stocks. The majority of the stocks with a mean $\lambda_{real}$ value less than one were in the central or NW Atlantic (N = 8), and included 5 of the 14 Atlantic cod (*G. morhua*) stocks.

Several species are harvested throughout the North Atlantic, including two relatively large and long-lived Gadidae species: Haddock (*Melanogrammus aeglefinus*) and Atlantic cod (*Gadus morhua*). These two species exhibited some of the highest variability in population growth rates both over time and among stocks (Fig 2). For both species, the variability in the population growth rate declined moving from the Eastern Atlantic to the Western Atlantic (Fig 4). The pattern was most pronounced for Atlantic cod, where stocks in the Eastern Atlantic rarely experienced high values of $\lambda$ (Figs 2 and 4). Pollock (*Pollachius virens*) is another Gadidae species harvested across the North Atlantic. The variability in $\lambda$ was lower for this species throughout its range (Figs 2 and 4). Atlantic herring (*Clupea harengus*) is a relatively small fast-growing species also found throughout the North Atlantic. However, there were no clear regional patterns in $\lambda$ for this species and its variability in $\lambda$ was lower than observed in several Atlantic cod and haddock stocks.

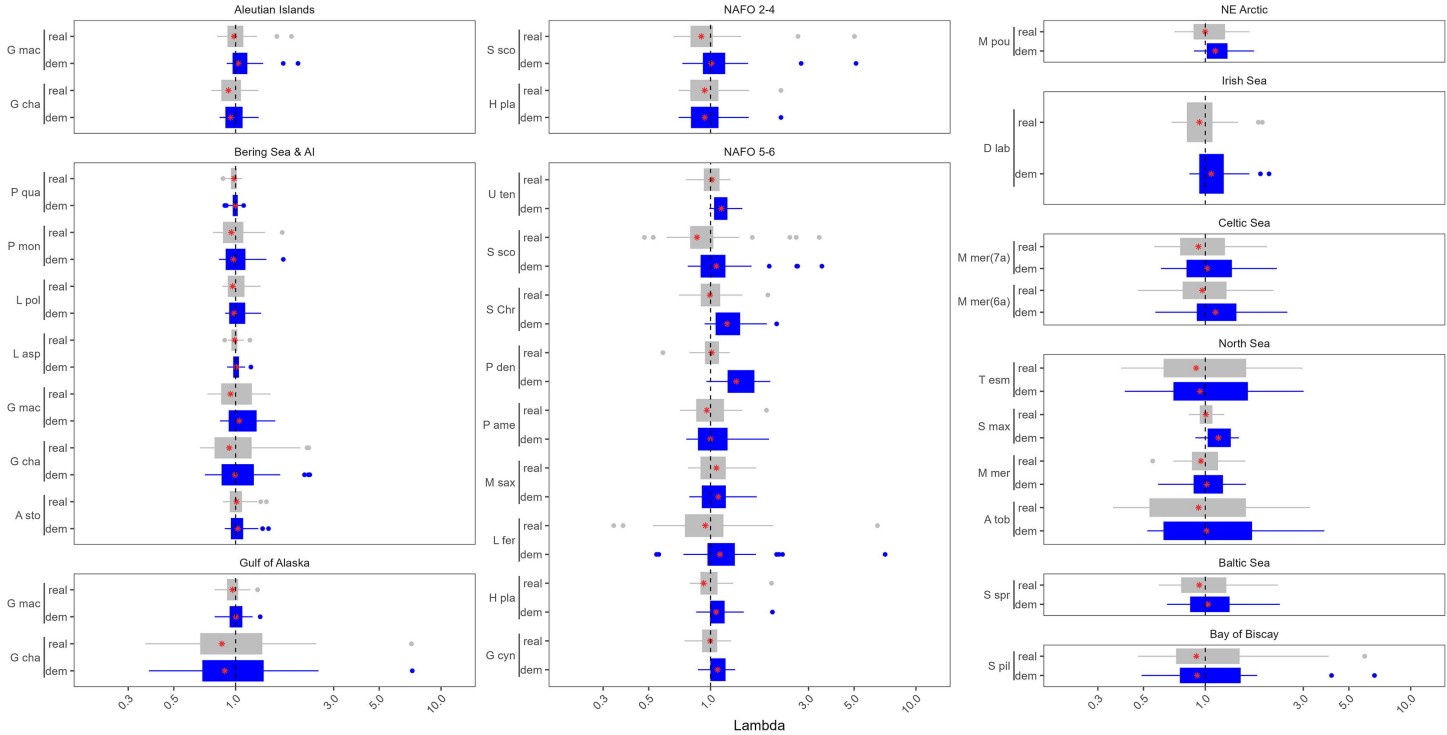

**Fig 3. Boxplot of the λ estimates for the remainder of the stocks in this analysis, plotted by the location of the stock.** The abbreviated species name is provided on the y axis. Blue are the $\lambda_{dem}$ estimates and grey are the $\lambda_{real}$ estimates. The red stars indicate the median and the dashed vertical line represents a λ of one. The x axis is on the log scale.

Two relatively short-lived and fast-growing species (Sand eel [*Ammodytes tobianus*] and Norway pout [*Trisopterus esmarkii*]) in the North Sea exhibited high variability in λ, though this variability was smaller than that observed in the haddock and Atlantic cod stocks in the area (Fig 3). For the remainder of the other stocks in this analysis, years in which the value of $\lambda_{dem}$ were elevated (e.g., above ≈ 1.5) were rarer, but no clear taxonomic or regional patterns were evident, with whiting (*Merlangus merlangus*) in the Celtic Sea, summer flounder (*Paralichthys dentatus*) and scup (*Stenotomus Chrysops*) in the Western Atlantic, and Gulf of Alaska pollock (*Gadus chalcogrammus*) all having moderate variability in λ (Fig 3).

There was no clear temporal trend in the lambda estimates, however $\lambda_{dem}$ in the 2000s was low (1.08), and in the 2010s the standard deviation of $\lambda_{dem}$ was higher than in the other decades, resulting in a low median $\lambda_{dem}$ (1.07), but a mean $\lambda_{dem}$ estimate similar to what was observed before the 2000s (1.23; Table 1). The high standard deviation in the 2010s was somewhat impacted by a single stock (CS RA *M. aeglefinus*). When this stock was removed from the analysis the standard deviation remained the highest in the time series, but was not as extreme (sd of $\lambda_{dem}$ declined to 0.63), while the median and mean λ's remained similar to values observed before the 2000s (e.g., the median and mean $\lambda_{dem}$ were 1.07 and 1.2 respectively). This analysis also indicated that relative abundances (proportion of maximum observed) of the stocks have been in decline over the last 40 years, with the lowest values observed in the 2010s (Table 1).

## Density dependence

Overall, both $\lambda_{dem}$ and $\lambda_{real}$ increased as abundance declined. The changes in the rate of population growth were more pronounced at low and high abundance levels. The relationship between λ and the proportion of maximum abundance

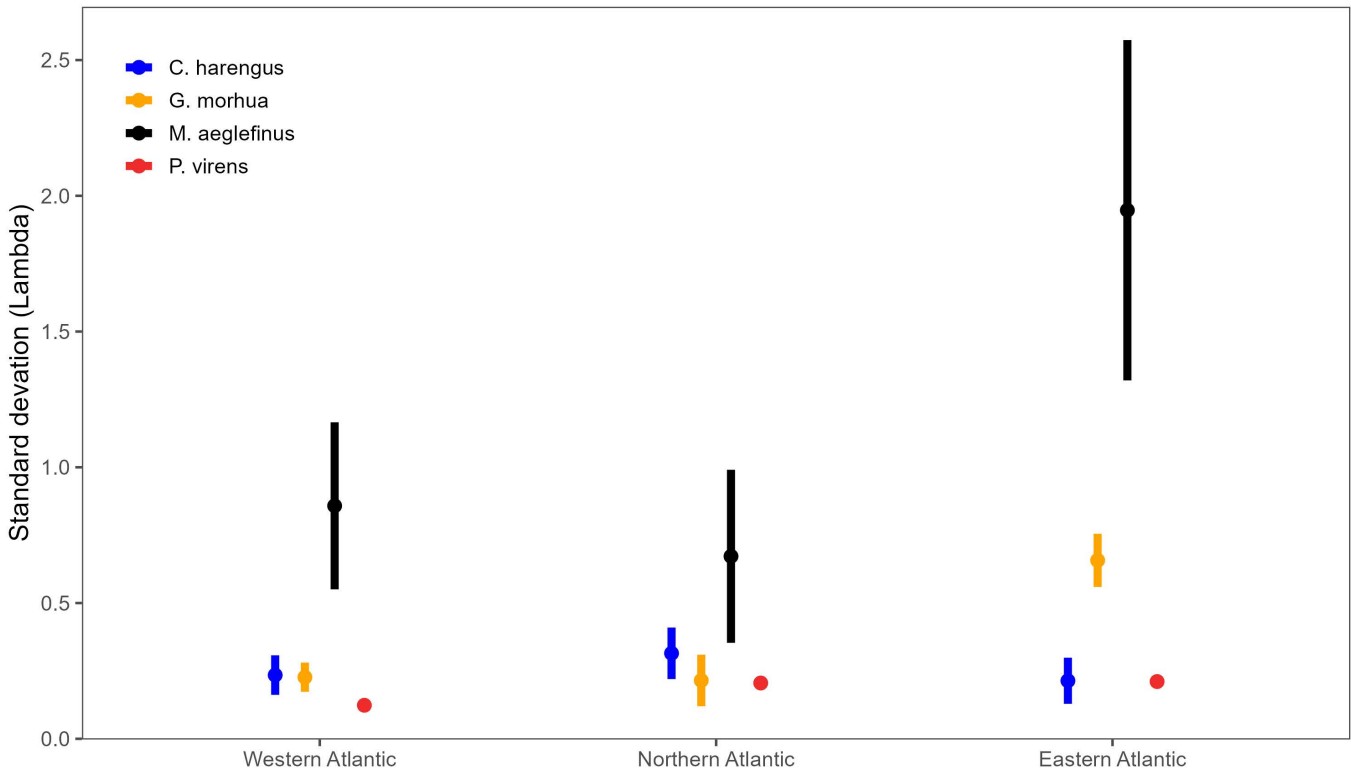

**Fig 4. The average standard deviation of $\lambda_{dem}$ for Atlantic herring (*C. harengus*), Atlantic cod (*G. morhua*), haddock (*M. aeglefinus*), and pollock (\*P. virens) stocks in the western Atlantic (NAFO regions 2-5), Northern Atlantic (Iceland, Faroe's, and the northeast Arctic), and Eastern Atlantic (Irish, Celtic, North, and Baltic Seas). The error bars represent one standard deviation.**

**Table 1. A comparison of the median, mean, and standard deviation of $\lambda_{dem}$ and $\lambda_{real}$ by decade for all the stocks in this analysis. The table includes the number of stocks included in each decade (N) and the median proporiton of the maximum observed abundance (Prop of max).**

| Decade | $\lambda_{dem}^{Median}$ | $r_{dem}^{Median}$ | $\lambda_{dem}^{Mean}$ | $\lambda_{dem}^{SD}$ | $\lambda_{real}^{Median}$ | $r_{real}^{Median}$ | $\lambda_{real}^{Mean}$ | $\lambda_{real}^{SD}$ | N | Prop of max |
|---|---|---|---|---|---|---|---|---|---|---|
| 1940 | 1.08 | 0.076 | 1.08 | 0.14 | 1.03 | 0.034 | 1.03 | 0.14 | 1 | 0.87 |
| 1950 | 1.04 | 0.035 | 1.11 | 0.29 | 0.95 | −0.050 | 0.99 | 0.27 | 8 | 0.65 |
| 1960 | 1.15 | 0.143 | 1.20 | 0.34 | 1.01 | 0.012 | 1.03 | 0.31 | 13 | 0.50 |
| 1970 | 1.10 | 0.093 | 1.22 | 0.50 | 0.95 | −0.053 | 1.04 | 0.46 | 39 | 0.50 |
| 1980 | 1.10 | 0.091 | 1.21 | 0.53 | 0.97 | −0.032 | 1.05 | 0.49 | 69 | 0.55 |
| 1990 | 1.11 | 0.103 | 1.21 | 0.56 | 0.98 | −0.020 | 1.05 | 0.48 | 76 | 0.48 |
| 2000 | 1.08 | 0.073 | 1.15 | 0.43 | 0.96 | −0.037 | 1.02 | 0.39 | 77 | 0.45 |
| 2010 | 1.07 | 0.069 | 1.23 | 0.92 | 0.99 | −0.012 | 1.12 | 0.89 | 77 | 0.42 |

was relatively flat when abundance ranged between 40–70% of maximum (Fig 5). The $\lambda_{dem}$ remained above one when stocks approached their maximum observed abundance in more than 87% of the stocks (Tables 2 and 3 and S1 and S2 Figs). However, $\lambda_{real}$ dropped below one when the stock abundance exceeded approximately 33% of the maximum abundance. At lower abundances, the model estimated $\lambda_{dem}$ increased as abundance declined more slowly than $\lambda_{real}$ (Fig 5). This is likely due to reductions in fishing mortality at low abundance, but since the fitted values did not overlap, some

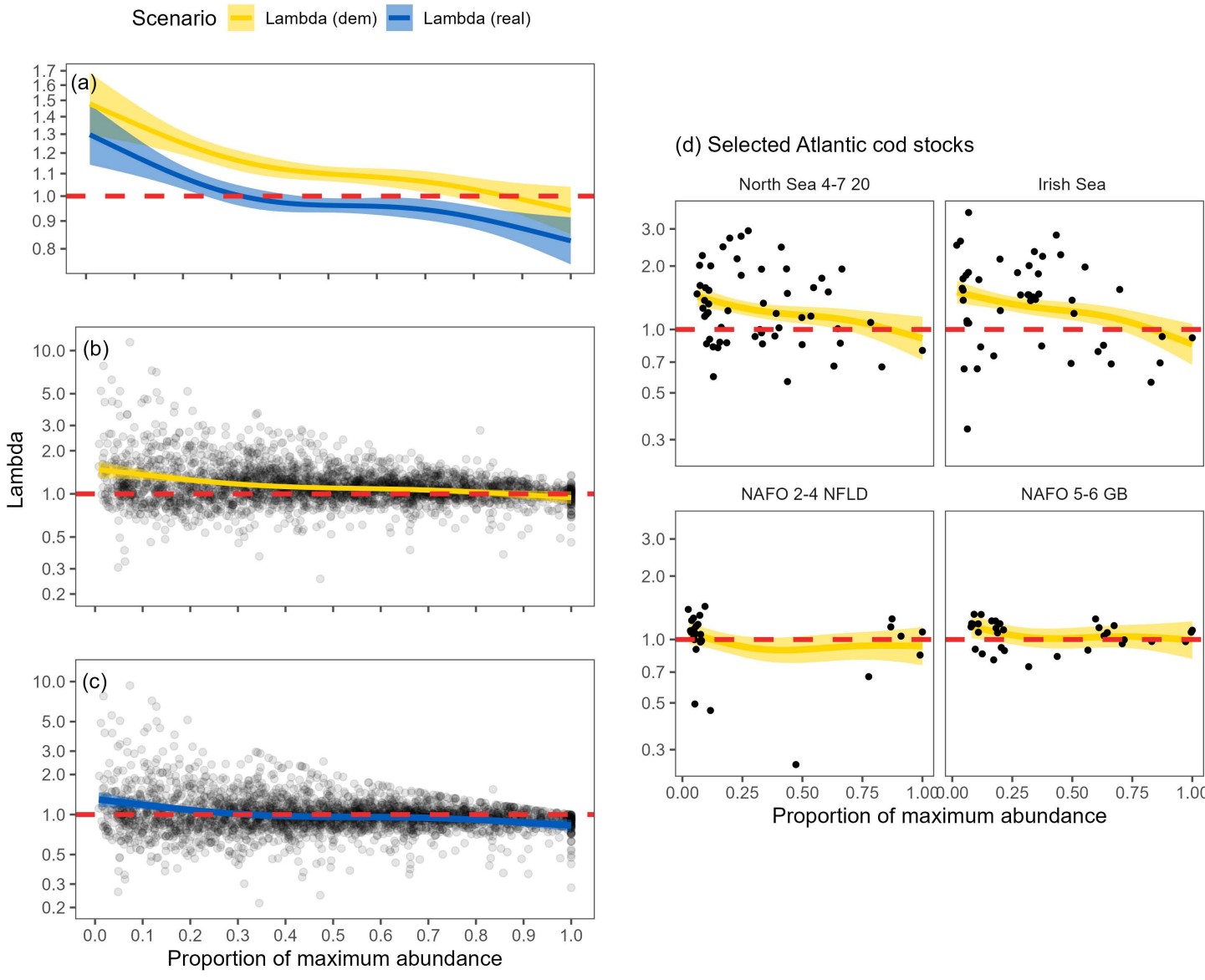

**Fig 5. Density dependence of λ.** Panel (a) compares the GAM smooths for $\lambda_{dem}$ and $\lambda_{real}$ and the GAM smooth and the GAM estimated 95% confidence interval. Panel (b) shows $\lambda_{dem}$ against the proportion of maximum abundance. The points represent the estimate for each year and stock along with the GAM smooth and the GAM estimated 95% confidence interval. Panel (c) shows $\lambda_{real}$ against the proportion of maximum abundance. Panel (d) shows the data and GAM smooths for two Atlantic cod (*G. morhua*) stocks in the Northeast Atlantic and two stocks in the Northwest Atlantic.

fishing mortality remained even at very low abundance. For the vast majority of stocks, the lowest model-estimated values of $\lambda_{dem}$ (75 of 77) were found at the maximum abundance in the time series (Tables 2 and 3 and S1 and S2 Figs). However, two Atlantic cod stocks in the Northwest Atlantic had their lowest population growth rates observed at abundances markedly below the maximum. These Atlantic cod stocks seemed to have the least compensatory capacity at relative abundances near 40% of their maximum.

**Table 2. A comparison of the $\lambda_{dem}$ and $\lambda_{real}$ estimates for stock at the minimum and maximum abundance, along with the estimate at 40% of the maximum abundance. Values greater than 1 indicate that the population would tend to increase in size, while values less than 1 indicate the population would tend to decline in size.**

| Stock | Minimum Abundance | $\lambda_{real}$ Min | 40% | Max | $\lambda_{dem}$ Min | 40% | Max | Species |
|---|---|---|---|---|---|---|---|---|
| NAFO 5-6 S. Chrysops | 0.12 | 1.130 | 0.986 | 0.909 | 1.526 | 1.231 | 0.961 | S. Chrysops |
| CS RA M. aeglefinus | 0.02 | 1.258 | 0.931 | 0.789 | 1.482 | 1.135 | 1.013 | M. aeglefinus |
| IS M. aeglefinus | 0.10 | 1.251 | 0.926 | 0.662 | 1.623 | 1.127 | 0.779 | M. aeglefinus |
| Ice-Faroe 5b M. aeglefinus | 0.11 | 1.161 | 0.974 | 0.867 | 1.282 | 1.102 | 1.066 | M. aeglefinus |
| Ice-Faroe 5a M. aeglefinus | 0.16 | 1.126 | 0.976 | 0.827 | 1.278 | 1.119 | 0.961 | M. aeglefinus |
| NAFO 5-6 GB M. aeglefinus | 0.01 | 1.185 | 0.933 | 0.837 | 1.326 | 1.049 | 0.920 | M. aeglefinus |
| NAFO 5-6 GOM M. aeglefinus | 0.04 | 1.211 | 0.969 | 0.841 | 1.354 | 1.095 | 0.937 | M. aeglefinus |
| NEA M. aeglefinus | 0.02 | 1.211 | 0.903 | 0.789 | 1.386 | 0.984 | 0.869 | M. aeglefinus |
| NS M. aeglefinus | 0.02 | 1.265 | 0.820 | 0.547 | 1.598 | 1.054 | 0.694 | M. aeglefinus |
| NAFO 5-6 P. americanus | 0.27 | 1.072 | 1.012 | 0.864 | 1.193 | 1.108 | 0.908 | P. americanus |
| BSAI L. aspera | 0.63 | 1.001 | NA | 0.906 | 1.045 | NA | 0.934 | L. aspera |
| AI G. chalcogrammus | 0.12 | 1.051 | 0.917 | 0.850 | 1.075 | 0.965 | 0.918 | G. chalcogrammus |
| ESB G. chalcogrammus | 0.29 | 1.165 | 1.079 | 0.754 | 1.298 | 1.191 | 0.793 | G. chalcogrammus |
| GOA G. chalcogrammus | 0.07 | 1.230 | 0.897 | 0.671 | 1.292 | 0.926 | 0.708 | G. chalcogrammus |
| NAFO 5-6 G. cynoglossus | 0.31 | 1.015 | 0.991 | 0.896 | 1.144 | 1.114 | 1.012 | G. cynoglossus |
| NAFO 5-6 P. dentatus | 0.33 | 1.033 | 1.014 | 0.880 | 1.450 | 1.419 | 1.135 | P. dentatus |
| NS T. esmarkii | 0.08 | 1.467 | 1.014 | 0.632 | 1.594 | 1.109 | 0.674 | T. esmarkii |
| NAFO 5-6 L. ferruginea | 0.03 | 1.122 | 0.888 | 0.785 | 1.307 | 1.077 | 0.907 | L. ferruginea |
| BS 22-24 C. harengus | 0.36 | 0.992 | 0.982 | 0.858 | 1.190 | 1.182 | 1.147 | C. harengus |
| BS 28 C. harengus | 0.23 | 1.133 | 1.043 | 0.862 | 1.316 | 1.229 | 1.014 | C. harengus |
| BS 25-32 C. harengus | 0.50 | 1.012 | NA | 0.910 | 1.153 | NA | 0.990 | C. harengus |
| CS C. harengus | 0.22 | 1.071 | 0.986 | 0.848 | 1.139 | 1.060 | 0.947 | C. harengus |
| Ice-Faroe C. harengus | 0.38 | 1.001 | 0.997 | 0.884 | 1.143 | 1.136 | 0.975 | C. harengus |
| NAFO 2-4 Fa-SGSL C. harengus | 0.41 | 1.029 | NA | 0.900 | 1.160 | NA | 0.976 | C. harengus |
| NAFO 2-4 Sp-SGSL C. harengus | 0.12 | 1.092 | 0.957 | 0.867 | 1.221 | 1.084 | 0.951 | C. harengus |
| NEA C. harengus | 0.24 | 1.149 | 1.038 | 0.781 | 1.259 | 1.131 | 0.837 | C. harengus |
| NS C. harengus | 0.06 | 1.221 | 0.980 | 0.839 | 1.496 | 1.188 | 0.927 | C. harengus |
| IS D. labrax | 0.06 | 1.121 | 0.938 | 0.852 | 1.256 | 1.082 | 0.958 | D. labrax |
| AI G. macrocephalus | 0.18 | 1.123 | 1.015 | 0.902 | 1.237 | 1.121 | 0.987 | G. macrocephalus |
| BSAI G. macrocephalus | 0.39 | 1.031 | 1.029 | 0.826 | 1.153 | 1.152 | 0.902 | G. macrocephalus |
| GOA G. macrocephalus | 0.43 | 0.998 | NA | 0.860 | 1.084 | NA | 0.889 | G. macrocephalus |
| NS S. maximus | 0.60 | 1.012 | NA | 0.911 | 1.164 | NA | 1.075 | S. maximus |
| CS 27.6a M. merlangus | 0.05 | 1.137 | 0.937 | 0.845 | 1.308 | 1.118 | 0.999 | M. merlangus |
| CS 7a M. merlangus | 0.11 | 1.074 | 0.926 | 0.842 | 1.182 | 1.049 | 0.929 | M. merlangus |
| NS M. merlangus | 0.34 | 1.032 | 1.010 | 0.828 | 1.125 | 1.096 | 0.884 | M. merlangus |
| BSAI P. monopterygius | 0.41 | 1.005 | NA | 0.829 | 1.073 | NA | 0.866 | P. monopterygius |
| BS G. morhua | 0.10 | 1.190 | 0.952 | 0.789 | 1.585 | 1.316 | 1.204 | G. morhua |
| CS G. morhua | 0.11 | 1.278 | 0.958 | 0.692 | 1.665 | 1.232 | 0.896 | G. morhua |
| IS G. morhua | 0.02 | 1.129 | 0.898 | 0.721 | 1.503 | 1.252 | 0.850 | G. morhua |
| Ice-Faroe 5b G. morhua | 0.12 | 1.136 | 0.972 | 0.842 | 1.317 | 1.156 | 0.973 | G. morhua |
| Ice-Faroe 5a G. morhua | 0.47 | 1.004 | NA | 0.905 | 1.135 | NA | 1.036 | G. morhua |
| NAFO 2-4 NFLD G. morhua | 0.03 | 1.055 | 0.860 | 0.838 | 1.072 | 0.890 | 0.931 | G. morhua |
| NAFO 2-4 NGSL G. morhua | 0.06 | 1.033 | 0.888 | 0.901 | 1.078 | 0.952 | 0.999 | G. morhua |

*(Continued)*

**Table 2.** (Continued)

| Stock | Minimum Abundance | Min | $\lambda_{real}$ 40% | Max | Min | $\lambda_{dem}$ 40% | Max | Species |
|---|---|---|---|---|---|---|---|---|
| *NAFO 2-4 SGSL G. morhua* | 0.07 | 1.110 | 0.941 | 0.874 | 1.109 | 0.985 | 0.934 | *G. morhua* |
| *NAFO 5-6 EGB G. morhua* | 0.14 | 1.072 | 0.940 | 0.862 | 1.166 | 1.055 | 1.012 | *G. morhua* |
| *NAFO 5-6 GB G. morhua* | 0.08 | 1.055 | 0.904 | 0.870 | 1.156 | 1.011 | 0.993 | *G. morhua* |
| *NAFO 5-6 GOM G. morhua* | 0.22 | 1.024 | 0.949 | 0.832 | 1.238 | 1.166 | 1.055 | *G. morhua* |
| *NEA G. morhua* | 0.10 | 1.174 | 0.958 | 0.800 | 1.346 | 1.098 | 0.887 | *G. morhua* |
| *NS 6a G. morhua* | 0.04 | 1.112 | 0.896 | 0.760 | 1.405 | 1.183 | 0.943 | *G. morhua* |
| *NS 4-7 20 G. morhua* | 0.06 | 1.173 | 0.912 | 0.713 | 1.432 | 1.196 | 0.907 | *G. morhua* |
| *BoB S. pilchardus* | 0.17 | 1.257 | 1.037 | 0.748 | 1.452 | 1.170 | 0.778 | *S. pilchardus* |
| *IS 7a P. platessa* | 0.38 | 1.009 | 1.006 | 0.919 | 1.258 | 1.250 | 1.033 | *P. platessa* |
| *IS 7e P. platessa* | 0.25 | 1.072 | 1.012 | 0.893 | 1.314 | 1.233 | 1.048 | *P. platessa* |
| *NS 4 20 P. platessa* | 0.24 | 1.097 | 1.011 | 0.847 | 1.433 | 1.320 | 1.099 | *P. platessa* |
| *NS 7d P. platessa* | 0.22 | 1.092 | 1.002 | 0.862 | 1.219 | 1.114 | 0.946 | *P. platessa* |
| *NAFO 2-4 SGSL H. platessoides* | 0.20 | 1.105 | 0.979 | 0.795 | 1.146 | 1.008 | 0.826 | *H. platessoides* |
| *NAFO 5-6 H. platessoides* | 0.36 | 1.030 | 1.018 | 0.838 | 1.159 | 1.145 | 0.985 | *H. platessoides* |
| *BSAI L. polyxystra* | 0.37 | 1.020 | 1.014 | 0.907 | 1.097 | 1.087 | 0.941 | *L. polyxystra* |
| *NEA M. poutassou* | 0.17 | 1.119 | 0.991 | 0.861 | 1.249 | 1.126 | 1.010 | *M. poutassou* |
| *BSAI P. quadrituberculatus* | 0.45 | 0.982 | NA | 0.902 | 1.047 | NA | 0.930 | *P. quadrituberculatus* |
| *NAFO 5-6 M. saxatilis* | 0.24 | 1.128 | 1.049 | 0.883 | 1.213 | 1.126 | 0.908 | *M. saxatilis* |
| *NAFO 2-4 SGSL S. scombrus* | 0.13 | 1.154 | 0.960 | 0.804 | 1.362 | 1.095 | 0.883 | *S. scombrus* |
| *NAFO 5-6 S. scombrus* | 0.02 | 1.071 | 0.859 | 0.806 | 1.256 | 0.981 | 0.906 | *S. scombrus* |
| *CS S. solea* | 0.47 | 1.008 | NA | 0.879 | 1.197 | NA | 1.037 | *S. solea* |
| *IS S. solea* | 0.50 | 1.011 | NA | 0.908 | 1.166 | NA | 1.055 | *S. solea* |
| *NS 22-24 S. solea* | 0.25 | 1.017 | 0.967 | 0.876 | 1.172 | 1.118 | 1.041 | *S. solea* |
| *NS 7d S. solea* | 0.52 | 1.017 | NA | 0.915 | 1.227 | NA | 1.113 | *S. solea* |
| *NS 4 S. solea* | 0.27 | 1.121 | 1.031 | 0.788 | 1.423 | 1.291 | 0.954 | *S. solea* |
| *BS S. sprattus* | 0.12 | 1.168 | 0.973 | 0.812 | 1.229 | 1.059 | 0.917 | *S. sprattus* |
| *BSAI A. stomais* | 0.22 | 1.104 | 1.027 | 0.918 | 1.194 | 1.097 | 0.951 | *A. stomais* |
| *NAFO 5-6 U. tenuis* | 0.29 | 1.055 | 1.013 | 0.878 | 1.193 | 1.147 | 1.051 | *U. tenuis* |
| *NS A. tobianus* | 0.14 | 1.304 | 0.942 | 0.565 | 1.495 | 1.054 | 0.644 | *A. tobianus* |
| *Ice-Faroe 5a P. virens* | 0.32 | 1.032 | 1.005 | 0.859 | 1.184 | 1.146 | 0.955 | *P. virens* |
| *Ice-Faroe 5b P. virens* | 0.25 | 1.067 | 1.001 | 0.873 | 1.217 | 1.129 | 0.959 | *P. virens* |
| *NAFO 5-6 P. virens* | 0.29 | 1.041 | 1.004 | 0.875 | 1.173 | 1.121 | 0.932 | *P. virens* |
| *NEA P. virens* | 0.23 | 1.086 | 1.006 | 0.861 | 1.347 | 1.223 | 0.951 | *P. virens* |
| *NS P. virens* | 0.29 | 1.052 | 0.994 | 0.791 | 1.272 | 1.207 | 0.998 | *P. virens* |

There was generally an increase in the variability of the $\lambda$ estimates as abundance declined, largely due to the increase in the magnitude of the maximum value of $\lambda$ (Fig 5). While there was a consistent overall pattern, the estimated random effects demonstrated substantial variability among stocks via the strength of negative density dependence. For example, stocks in the Western Atlantic (particularly Atlantic cod) and Alaska had some of the lowest $\lambda_{dem}$ estimates at low abundance, while several stocks in the North Sea (including Atlantic cod) had among the highest $\lambda_{dem}$ estimates (Table 2). Of the 66 stocks that declined below 40% of the maximum abundance, 7 had $\lambda_{dem}$ estimates < 1 at 40% of their maximum abundance; when the effects of fishing were included, 42 of the stocks had $\lambda_{real}$ below 1 (Table 2).

**Table 3. A comparison of the difference in $\lambda_{dem}$ and $\lambda_{real}$ for each stock at the minimum and maximum abundance, along with the change at 40% of the maximum abundance. Positive values indicate that the $\lambda$ estimate has increased as abundance declines (consistent with negative density dependence), while negative values indicate the $\lambda$ estimate has declined (consistent with positive density dependence).**

| Stock | Minimum Abundance | Min-40 | $\lambda_{real}$ Min-Max | 40-Max | Min-40 | $\lambda_{dem}$ Min-Max | 40-Max | Species |
|---|---|---|---|---|---|---|---|---|
| NAFO 5-6 S. Chrysops | 0.12 | 0.144 | 0.221 | 0.077 | 0.295 | 0.564 | 0.270 | S. Chrysops |
| CS RA M. aeglefinus | 0.02 | 0.328 | 0.469 | 0.141 | 0.347 | 0.469 | 0.122 | M. aeglefinus |
| IS M. aeglefinus | 0.10 | 0.326 | 0.589 | 0.263 | 0.496 | 0.844 | 0.348 | M. aeglefinus |
| Ice-Faroe 5b M. aeglefinus | 0.11 | 0.187 | 0.294 | 0.107 | 0.180 | 0.216 | 0.036 | M. aeglefinus |
| Ice-Faroe 5a M. aeglefinus | 0.16 | 0.150 | 0.298 | 0.149 | 0.159 | 0.317 | 0.158 | M. aeglefinus |
| NAFO 5-6 GB M. aeglefinus | 0.01 | 0.252 | 0.347 | 0.096 | 0.277 | 0.405 | 0.128 | M. aeglefinus |
| NAFO 5-6 GOM M. aeglefinus | 0.04 | 0.242 | 0.369 | 0.128 | 0.259 | 0.416 | 0.158 | M. aeglefinus |
| NEA M. aeglefinus | 0.02 | 0.308 | 0.422 | 0.114 | 0.403 | 0.518 | 0.115 | M. aeglefinus |
| NS M. aeglefinus | 0.02 | 0.445 | 0.718 | 0.272 | 0.544 | 0.904 | 0.360 | M. aeglefinus |
| NAFO 5-6 P. americanus | 0.27 | 0.061 | 0.208 | 0.148 | 0.084 | 0.285 | 0.200 | P. americanus |
| BSAI L. aspera | 0.63 | NA | 0.095 | NA | NA | 0.111 | NA | L. aspera |
| AI G. chalcogrammus | 0.12 | 0.134 | 0.201 | 0.067 | 0.109 | 0.157 | 0.047 | G. chalcogrammus |
| ESB G. chalcogrammus | 0.29 | 0.087 | 0.411 | 0.324 | 0.107 | 0.505 | 0.398 | G. chalcogrammus |
| GOA G. chalcogrammus | 0.07 | 0.332 | 0.559 | 0.227 | 0.366 | 0.584 | 0.218 | G. chalcogrammus |
| NAFO 5-6 G. cynoglossus | 0.31 | 0.023 | 0.119 | 0.096 | 0.031 | 0.132 | 0.102 | G. cynoglossus |
| NAFO 5-6 P. dentatus | 0.33 | 0.019 | 0.153 | 0.134 | 0.031 | 0.315 | 0.284 | P. dentatus |
| NS T. esmarkii | 0.08 | 0.453 | 0.835 | 0.381 | 0.485 | 0.920 | 0.435 | T. esmarkii |
| NAFO 5-6 L. ferruginea | 0.03 | 0.234 | 0.337 | 0.104 | 0.231 | 0.400 | 0.170 | L. ferruginea |
| BS 22-24 C. harengus | 0.36 | 0.010 | 0.133 | 0.123 | 0.007 | 0.043 | 0.036 | C. harengus |
| BS 28 C. harengus | 0.23 | 0.089 | 0.271 | 0.181 | 0.087 | 0.302 | 0.215 | C. harengus |
| BS 25-32 C. harengus | 0.50 | NA | 0.102 | NA | NA | 0.163 | NA | C. harengus |
| CS C. harengus | 0.22 | 0.084 | 0.223 | 0.139 | 0.079 | 0.192 | 0.112 | C. harengus |
| Ice-Faroe C. harengus | 0.38 | 0.004 | 0.118 | 0.114 | 0.007 | 0.168 | 0.161 | C. harengus |
| NAFO 2-4 Fa-SGSL C. harengus | 0.41 | NA | 0.129 | NA | NA | 0.184 | NA | C. harengus |
| NAFO 2-4 Sp-SGSL C. harengus | 0.12 | 0.135 | 0.224 | 0.090 | 0.137 | 0.270 | 0.132 | C. harengus |
| NEA C. harengus | 0.24 | 0.111 | 0.368 | 0.257 | 0.128 | 0.421 | 0.294 | C. harengus |
| NS C. harengus | 0.06 | 0.241 | 0.381 | 0.140 | 0.308 | 0.568 | 0.260 | C. harengus |
| IS D. labrax | 0.06 | 0.183 | 0.269 | 0.086 | 0.174 | 0.299 | 0.125 | D. labrax |
| AI G. macrocephalus | 0.18 | 0.108 | 0.221 | 0.113 | 0.117 | 0.250 | 0.134 | G. macrocephalus |
| BSAI G. macrocephalus | 0.39 | 0.001 | 0.205 | 0.204 | 0.002 | 0.252 | 0.250 | G. macrocephalus |
| GOA G. macrocephalus | 0.43 | NA | 0.138 | NA | NA | 0.195 | NA | G. macrocephalus |
| NS S. maximus | 0.60 | NA | 0.101 | NA | NA | 0.089 | NA | S. maximus |
| CS 27.6a M. merlangus | 0.05 | 0.201 | 0.292 | 0.091 | 0.190 | 0.309 | 0.119 | M. merlangus |
| CS 7a M. merlangus | 0.11 | 0.148 | 0.232 | 0.084 | 0.133 | 0.253 | 0.120 | M. merlangus |
| NS M. merlangus | 0.34 | 0.021 | 0.204 | 0.183 | 0.029 | 0.242 | 0.213 | M. merlangus |
| BSAI P. monopterygius | 0.41 | NA | 0.175 | NA | NA | 0.206 | NA | P. monopterygius |
| BS G. morhua | 0.10 | 0.238 | 0.401 | 0.163 | 0.269 | 0.381 | 0.112 | G. morhua |
| CS G. morhua | 0.11 | 0.320 | 0.586 | 0.266 | 0.433 | 0.768 | 0.335 | G. morhua |
| IS G. morhua | 0.02 | 0.231 | 0.408 | 0.177 | 0.251 | 0.653 | 0.402 | G. morhua |
| Ice-Faroe 5b G. morhua | 0.12 | 0.164 | 0.294 | 0.131 | 0.162 | 0.344 | 0.183 | G. morhua |
| Ice-Faroe 5a G. morhua | 0.47 | NA | 0.099 | NA | NA | 0.099 | NA | G. morhua |
| NAFO 2-4 NFLD G. morhua | 0.03 | 0.194 | 0.216 | 0.022 | 0.182 | 0.142 | -0.040 | G. morhua |
| NAFO 2-4 NGSL G. morhua | 0.06 | 0.144 | 0.132 | -0.012 | 0.126 | 0.079 | -0.047 | G. morhua |

*(Continued)*

**Table 3.** (Continued)

| Stock | Minimum Abundance | Min-40 | $\lambda_{real}$ Min-Max | 40-Max | Min-40 | $\lambda_{dem}$ Min-Max | 40-Max | Species |
|---|---|---|---|---|---|---|---|---|
| NAFO 2-4 SGSL G. morhua | 0.07 | 0.169 | 0.236 | 0.067 | 0.124 | 0.175 | 0.052 | G. morhua |
| NAFO 5-6 EGB G. morhua | 0.14 | 0.132 | 0.210 | 0.079 | 0.111 | 0.154 | 0.043 | G. morhua |
| NAFO 5-6 GB G. morhua | 0.08 | 0.151 | 0.186 | 0.035 | 0.144 | 0.162 | 0.018 | G. morhua |
| NAFO 5-6 GOM G. morhua | 0.22 | 0.076 | 0.193 | 0.117 | 0.072 | 0.183 | 0.111 | G. morhua |
| NEA G. morhua | 0.10 | 0.216 | 0.374 | 0.158 | 0.248 | 0.459 | 0.211 | G. morhua |
| NS 6a G. morhua | 0.04 | 0.216 | 0.352 | 0.136 | 0.222 | 0.462 | 0.240 | G. morhua |
| NS 4-7 20 G. morhua | 0.06 | 0.261 | 0.460 | 0.199 | 0.236 | 0.525 | 0.289 | G. morhua |
| BoB S. pilchardus | 0.17 | 0.220 | 0.509 | 0.289 | 0.283 | 0.674 | 0.391 | S. pilchardus |
| IS 7a P. platessa | 0.38 | 0.003 | 0.091 | 0.088 | 0.008 | 0.225 | 0.217 | P. platessa |
| IS 7e P. platessa | 0.25 | 0.059 | 0.179 | 0.119 | 0.081 | 0.266 | 0.185 | P. platessa |
| NS 4 20 P. platessa | 0.24 | 0.086 | 0.250 | 0.164 | 0.113 | 0.334 | 0.221 | P. platessa |
| NS 7d P. platessa | 0.22 | 0.090 | 0.230 | 0.140 | 0.105 | 0.274 | 0.169 | P. platessa |
| NAFO 2-4 SGSL H. platessoides | 0.20 | 0.127 | 0.311 | 0.184 | 0.137 | 0.320 | 0.183 | H. platessoides |
| NAFO 5-6 H. platessoides | 0.36 | 0.012 | 0.192 | 0.180 | 0.014 | 0.174 | 0.159 | H. platessoides |
| BSAI L. polyxystra | 0.37 | 0.005 | 0.113 | 0.107 | 0.010 | 0.156 | 0.146 | L. polyxystra |
| NEA M. poutassou | 0.17 | 0.128 | 0.258 | 0.130 | 0.122 | 0.239 | 0.116 | M. poutassou |
| BSAI P. quadrituberculatus | 0.45 | NA | 0.080 | NA | NA | 0.117 | NA | P. quadrituberculatus |
| NAFO 5-6 M. saxatilis | 0.24 | 0.079 | 0.246 | 0.166 | 0.087 | 0.305 | 0.218 | M. saxatilis |
| NAFO 2-4 SGSL S. scombrus | 0.13 | 0.195 | 0.350 | 0.156 | 0.267 | 0.479 | 0.212 | S. scombrus |
| NAFO 5-6 S. scombrus | 0.02 | 0.213 | 0.266 | 0.053 | 0.275 | 0.350 | 0.075 | S. scombrus |
| CS S. solea | 0.47 | NA | 0.129 | NA | NA | 0.160 | NA | S. solea |
| IS S. solea | 0.50 | NA | 0.103 | NA | NA | 0.110 | NA | S. solea |
| NS 22-24 S. solea | 0.25 | 0.050 | 0.141 | 0.091 | 0.054 | 0.132 | 0.078 | S. solea |
| NS 7d S. solea | 0.52 | NA | 0.102 | NA | NA | 0.114 | NA | S. solea |
| NS 4 S. solea | 0.27 | 0.091 | 0.333 | 0.242 | 0.131 | 0.469 | 0.338 | S. solea |
| BS S. sprattus | 0.12 | 0.195 | 0.355 | 0.160 | 0.170 | 0.312 | 0.142 | S. sprattus |
| BSAI A. stomais | 0.22 | 0.077 | 0.186 | 0.109 | 0.097 | 0.244 | 0.146 | A. stomais |
| NAFO 5-6 U. tenuis | 0.29 | 0.042 | 0.177 | 0.135 | 0.046 | 0.142 | 0.096 | U. tenuis |
| NS A. tobianus | 0.14 | 0.362 | 0.739 | 0.377 | 0.441 | 0.850 | 0.410 | A. tobianus |
| Ice-Faroe 5a P. virens | 0.32 | 0.028 | 0.173 | 0.145 | 0.037 | 0.229 | 0.191 | P. virens |
| Ice-Faroe 5b P. virens | 0.25 | 0.066 | 0.194 | 0.128 | 0.088 | 0.259 | 0.171 | P. virens |
| NAFO 5-6 P. virens | 0.29 | 0.037 | 0.166 | 0.129 | 0.052 | 0.241 | 0.189 | P. virens |
| NEA P. virens | 0.23 | 0.080 | 0.224 | 0.144 | 0.124 | 0.396 | 0.272 | P. virens |
| NS P. virens | 0.29 | 0.059 | 0.261 | 0.203 | 0.064 | 0.274 | 0.210 | P. virens |

## Recovery potential

When excluding fishing mortality, the overall median lifetime reproductive success (i.e., the number of female spawners per female spawner) was 2.8 (mean = 11.1; Figs 6 and S3 Fig). When fishing mortality was accounted for, the median reproductive success dropped to 1 (mean = 4.3). This difference between the mean and median for stocks experiencing exploitation again highlights the importance of occasional large reproductive events in sustaining these harvested stocks (Figs 6 and S3 Fig). For species found across the North Atlantic, the lifetime reproductive success when excluding fishing mortality tended to be higher in the east than the west, a pattern most evident in Atlantic cod (*G. morhua*) stocks. While

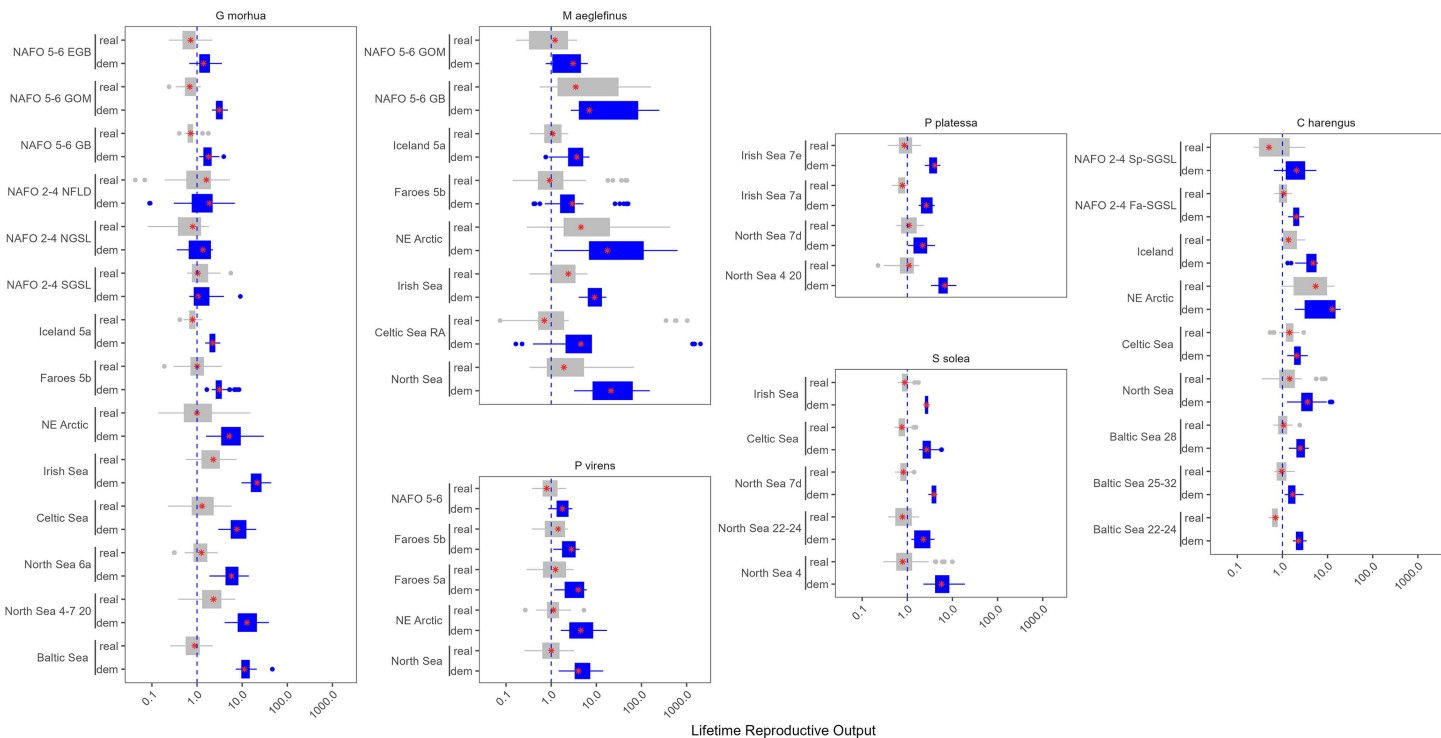

**Fig 6. R$_0$ Estimates for each stock of *G. morhua, M. aeglefinus, P. virens, C. harengus, P. platessa, and S. solea*.** The blue values are calculated using natural mortality and grey includes both natural mortality and fishing mortality. The red stars indicate the median and the dashed vertical line represents a lifetime reproductive output of one. The x axis is on the log scale.

the reproductive potential of cohorts within a stock could be highly variable, there were no clear temporal trends in lifetime reproductive success (S4 and S5 Figs).

Generation length (calculated as the mean age of spawners contributing to a recruit cohort) was lower and tended to be less variable under fishing across all stocks (Figs 7 and S6 Fig). Similar to other metrics, there were regional and inter-specific patterns. For example, Atlantic herring (*C. harengus*) is typically considered to be a short-lived species, yet its generation length in the Central and Western Atlantic tended to be higher than the generation length of Atlantic cod (*G. morhua*) and haddock (*M. aeglefinus*) in the Eastern Atlantic (Fig 7). Similar to lifetime reproductive output, there were no obvious temporal trends in the generation length among cohorts for the stocks in this analysis (S7 and S8 Figs).

Based on simulated stock trajectories, the doubling time in the absence of fishing was relatively high for the majority of species (Fig 8). Of the 77 stocks in this analysis, 39 were classified with a high recovery potential, which increased to 68 if those having a moderate recovery potential after 20 years were included (S9 Fig). Unlike stocks with higher recovery potential, there were clear regional patterns among stocks with low recovery potential. Of the 9 stocks in this analysis classified as having low recovery potential, 6 were in Alaska and 3 were in NAFO Regions 2–4 of the Western Atlantic. This meant that 55% and 43% of assessed stocks in Alaska and NAFO Regions 2–4, respectively, were classified as having a low recovery potential. This contrasts to 0 of the 44 stocks in the Eastern Atlantic classified with low recovery potential in the absence of fishing (Fig 8).

Doubling time was influenced by both the magnitude and variability of $\lambda_{dem}$. Stocks with a median $\lambda_{dem} < 1$ were classified as having moderate to high recovery potential whenever the mean $\lambda_{dem}$ was $\gtrsim 1.06$ or the standard deviation in $\lambda$ was

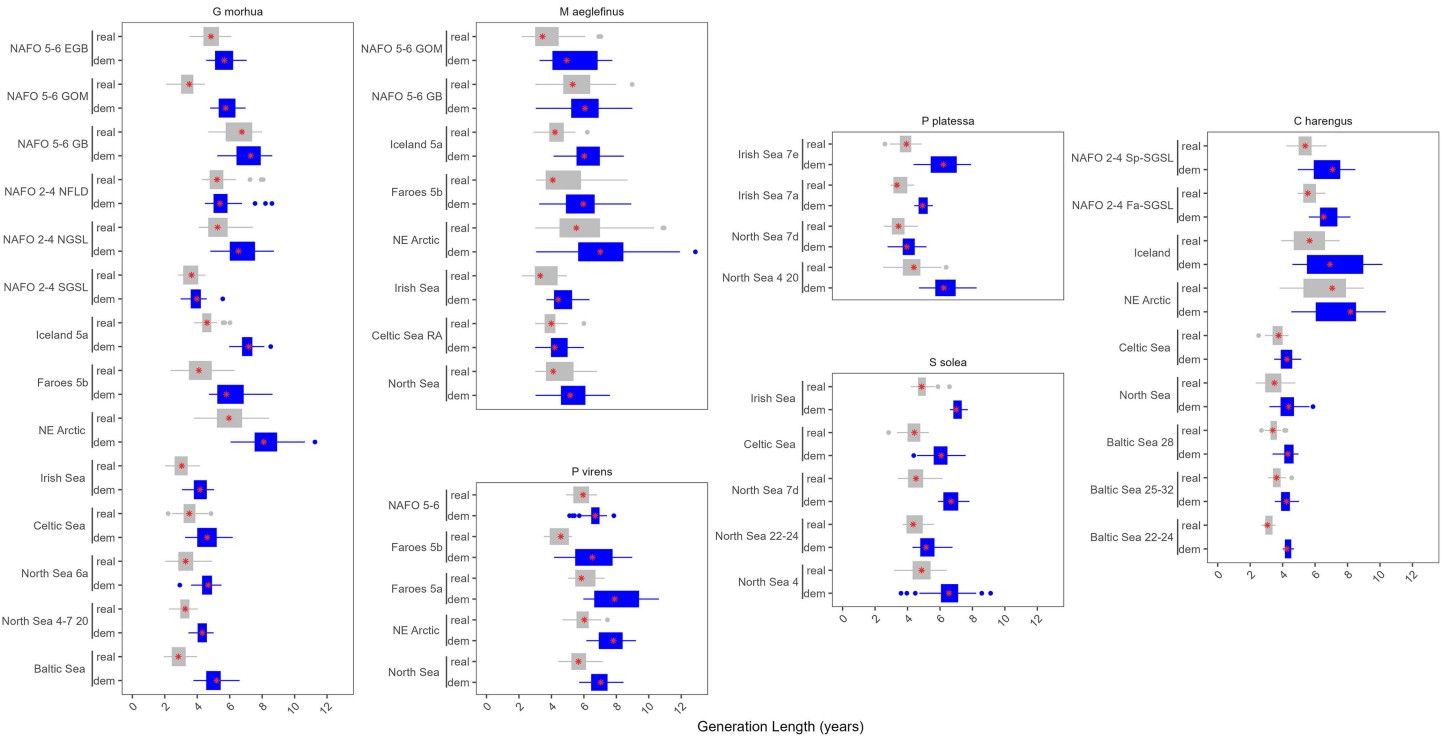

**Fig 7. Generation time estimates for each stock of *G. morhua, M. aeglefinus, P. virens, C. harengus, P. platessa,* and *S. solea*.** The blue values are calculated using natural mortality and grey includes both natural mortality and fishing mortality. The red stars indicate the median.

$\gtrsim$ 0.29 (Fig 9). In contrast, stocks with a median $\lambda_{dem}$ near 1 and with low variability tended to be classified as having low recovery potential.

## Discussion

By using life-tables to assess the recovery potential of 77 commercially harvested marine fish stocks, we found that most stocks have an innate capacity for population growth in the absence of fishing, with 88% displaying a moderate to high recovery potential. However, we found that these stocks declined in size in the majority of years (i.e., $\lambda_{real}$ was < 1 in 56.8% of the years) which suggests that stocks were managed in such a way that large but infrequent recruitment events were required to sustain productivity over the long term. Quota-based fisheries management strategies often implicitly or explicitly aim for neutral risk (i.e., a 50% probability) of population decline when setting harvest advice (e.g., [56]; [57]), however, this analysis indicates that this was not achieved in practice. In addition, there was considerable variability in the observed demographics and recovery potential within and among species and stocks. For example, while some Atlantic cod stocks exhibited strong negative density dependence and potential for recovery, other stocks demonstrated a limited capacity for population growth. Collectively, these findings highlight the importance of developing context-specific recovery strategies that consider the demographics of individual stocks, as a reliance on only life-history characteristics will likely prove insufficient to predict recovery potential.

Moreover, when the effects of fishing were excluded, these results indicated these stocks retained an innate capacity for increase of approximately 8.7%. If fishing ceased, the stocks were likely to recover (i.e., double in size) within 20 years whenever the median $\lambda_{dem}$ was $\gtrsim$ 1.06 or the standard deviation of lambda was $\gtrsim$ 0.29. Widespread distributions, high egg production per female, and broadcast spawning strategies suggest that harvested marine fishes should possess very

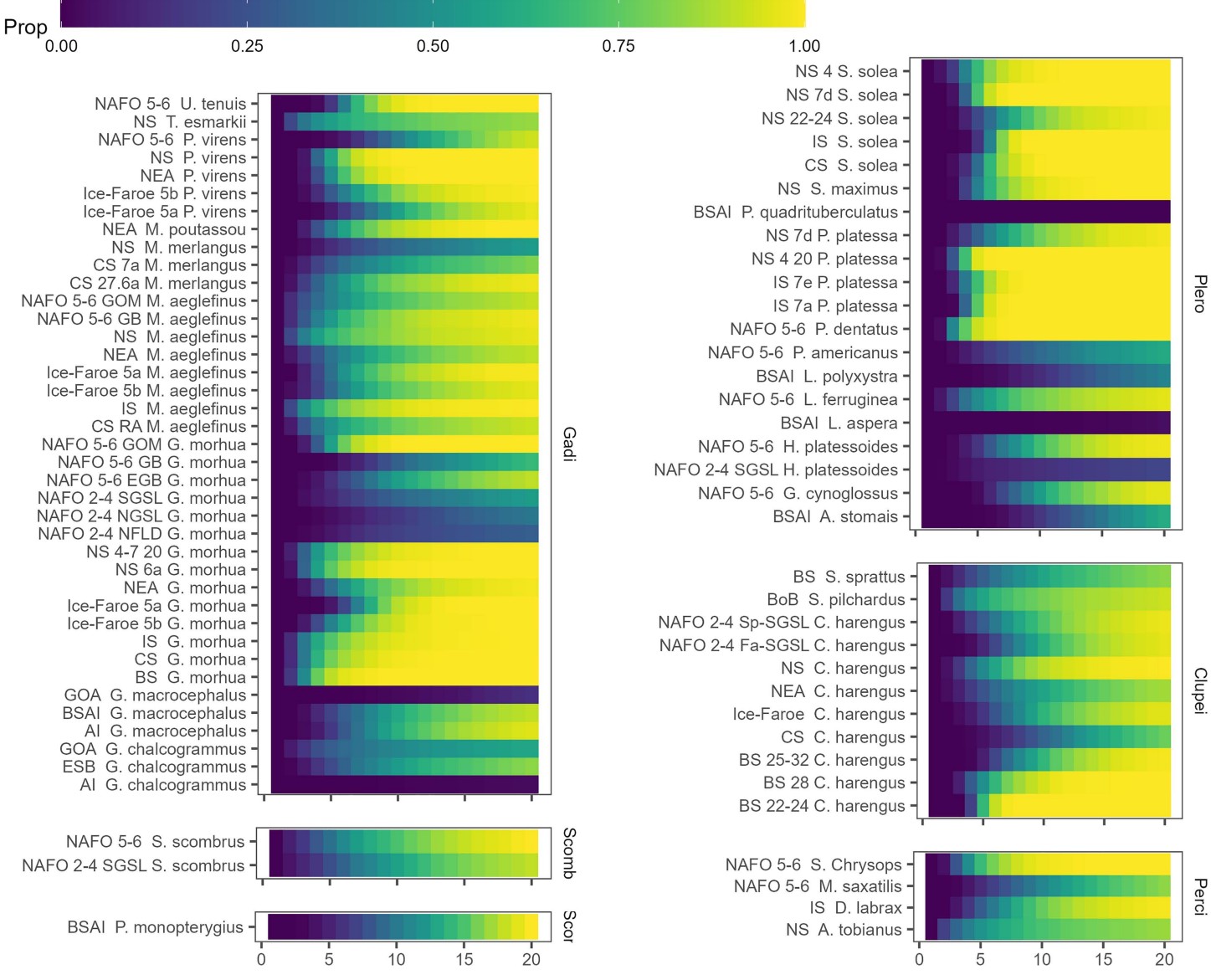

**Fig 8. Proportion of the simulations in which a given stock has doubled in size when there is no fishing mortality.** Stocks are grouped by taxonomic order, with 39 Gadiform stocks, 2 Scombriform stocks, 1 Scorpaeniform stock, 20 Pleuronectiform stocks, 11 Clupeiform stocks, and 4 Perciform stocks. The x-axis represents the year of the simulation, with the first 20 years shown.

high potential for annual population growth compared to other vertebrates [21]. Therefore, when environmental variation results in favorable conditions, stocks are anticipated to exhibit high survival, resulting in large recruitment events or higher overall survival (e.g., [58], [59]). While we found evidence of this, stocks with high variability in $\lambda$ also had high recovery potential, however, both intra- and inter-species variability did not conform to expectations from conventional life-history theory [25,60,61]. For example, Atlantic herring are a small, relatively short-lived, low-trophic-level pelagic species; their life-history characteristics are considered to result in highly variable population growth potential, which should allow them to exploit favorable environmental conditions and increase quickly in abundance [62,63]. However, the variability of $\lambda$ for several Atlantic herring stocks was lower than for the higher trophic level and longer-lived Atlantic cod

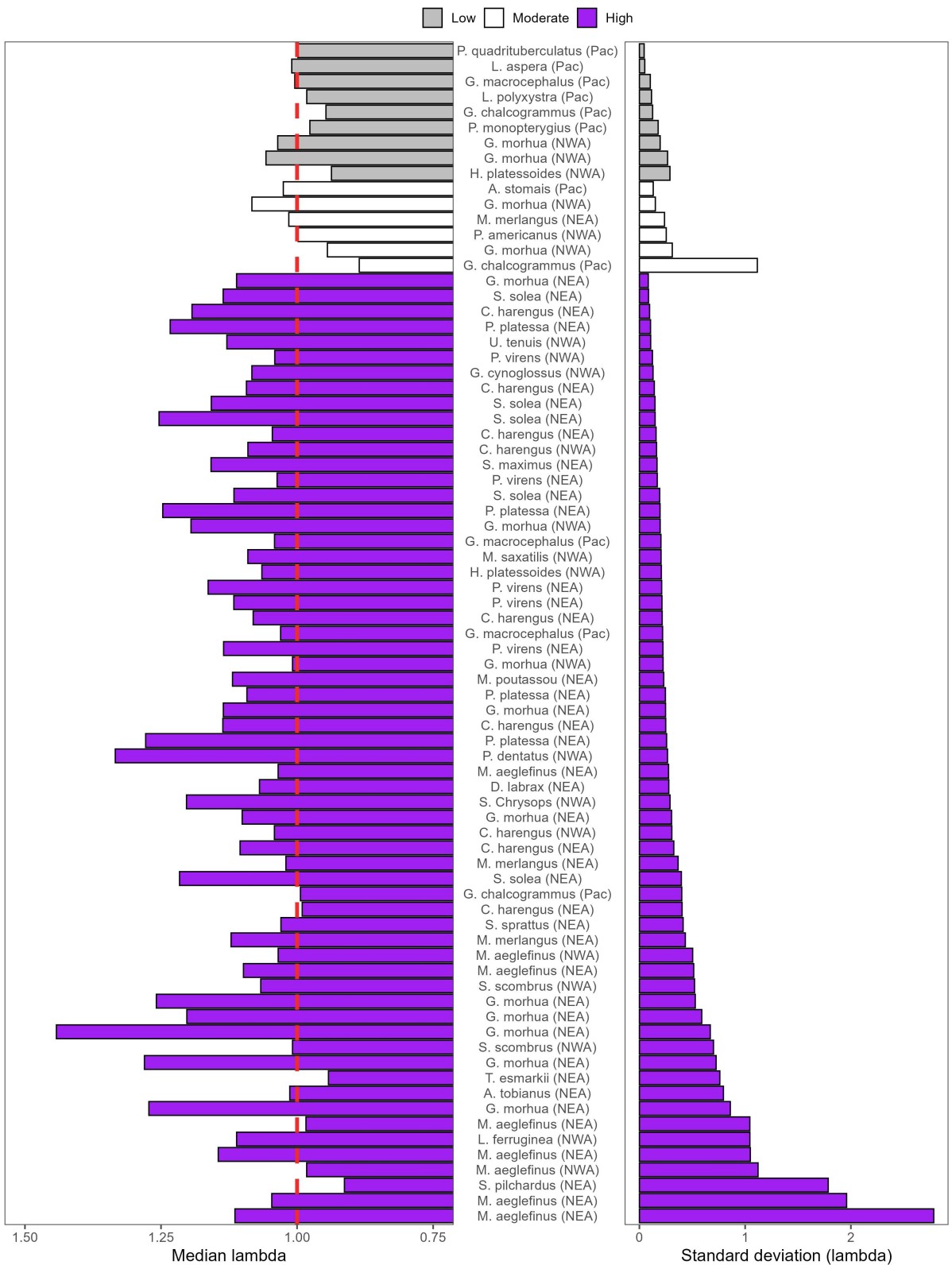

**Fig 9. The doubling time recovery potential, the colors represent low (grey), moderate (white), or high(blue) for each stock after 20 years compared to the median$\lambda_{dem}$ and the standard deviation of $\lambda_{dem}$. The red dashed line on the left panel is a $\lambda_{dem}$ value of one.**

and haddock stocks in the same region. Furthermore, these three species co-occur throughout the North Atlantic and the timespan of data collection was similar, suggesting they likely experienced similar variability in oceanographic conditions. Other stocks of small, lower-trophic-level species, such as lesser sand eel (*A. tobianus*) and Norway pout (*T. esmarkii*), were characterized by higher $\lambda$ values, which aligned with theoretical expectations for these species. However, these two stocks were also found in the North Sea, which appeared to be an environment in which high variability in $\lambda$ is commonplace for harvested stocks.

These results suggest that equilibrium theory and general life-history characteristics (e.g., longevity, trophic level) are poor predictors of population growth rates and recovery potential. For stocks that have collapsed or are considered to be at very low abundance, it would be prudent to evaluate the observed demographic parameters for the stock prior to developing harvest advice. Our analysis demonstrates that several stocks had little capacity to increase in abundance even in the absence of fishing, which would guarantee further declines if any level of harvesting continued. The stocks with limited recovery potential belonged to relatively distinct taxonomic and regional categories. For example, 7 stocks had $\lambda_{dem}$ estimates below one at 40% of their maximum abundance, indicating these stocks had limited capacity for population growth at low abundance in the absence of fishing. Six of these stocks were in the Gadidae family, with four in the NW Atlantic (three of which were Atlantic cod stocks) and two Alaskan pollock stocks (*G. chalcogrammus*). These stocks exhibited characteristics similar to endangered populations, which can also have lifetime reproductive rates < 1 [64,65].

In our analysis, the most marked example of intra-species variability in productivity was Atlantic cod across the North Atlantic. We have little understanding of how large most of these Atlantic cod stocks would have been before being impacted by large-scale harvesting, but historical reconstructions suggest the stocks in Iceland and the Western Atlantic were substantially larger [66–68]. Similar to many wild salmon populations [69,70], the combination of potential extreme abundance declines and enviornental changes may have resulted in these cod stocks having little innate capacity for growth or recovery. Indeed, it has been suggested that the Southern Gulf of St. Lawrence cod stock will go extinct unless natural mortality declines significantly [71,72], a conclusion supported by our analysis. However, the low recovery potential of Atlantic cod was not observed in the Eastern Atlantic; the stocks around mainland Europe all had high recovery potential. These regional differences may be related to fisheries-induced evolution, which has been linked to substantial change in the life-history characteristics (e.g., age and size at maturity), the loss of large old spawners, and/or to environmental conditions within the Western Atlantic [73–78]. This indicates that regional variability must factor strongly into a species' ability to withstand fishing pressure, irrespective of life-history. Given that variability in growth potential among populations could be substantially greater than among species (e.g., compare the variability of $\lambda$ between Atlantic cod and Atlantic herring), life-history characteristics and/or trophic level appear to be poor predictors of how marine fishes will respond to harvesting. This problem would compound for data-limited stocks, wherein these assessments often rely on species life-history traits (e.g., [79]). Consequently, such assumptions are more likely to lead to inappropriate harvest advice. That life-history characteristics did not link strongly to expected productivity is extremely concerning, given the need for sustainable fisheries management of data-limited stocks to mitigate risks of overfishing across the globe [79,80].

While no clear temporal trend was observed in $\lambda$, both the 2000s and 2010s were unusual relative to the previous three decades. In the decades with sufficient information to compare, the $\lambda_{dem}$ estimates were lowest in the 2000s and only the 1970s had a lower $\lambda_{real}$ when accounting for the effects of fishing. The median $\lambda_{dem}$ remained relatively low in the 2010s, however the variability of the $\lambda_{dem}$ estimates was higher than what had been observed previously, resulting in a mean $\lambda_{dem}$ similar to what was observed in earlier decades. Intriguingly, the abundance continued to decline relative to its maximum and reached the lowest level observed in the 2010s. It was approaching 40% of the maximum abundance, a level in which compensatory dynamics begin to be observed more clearly for many species [50]. Compensatory response should cause $\lambda_{dem}$ to increase. Thus, the increase in the variability in the 2010s may be related to stronger compensatory dynamics in some stocks (e.g., an increase in years with relatively large recruitment events). However, it is also possible that broad

scale climate effects could increase variability, given that phenological changes (e.g., timing of reproduction, physiology, behaviour) brought on by environmental variability are expected to affect demographic rates and thus population dynamics [81]. Finally, estimates of cohort size from age structured models are the most uncertain at the end of the time series, because the most recent cohorts have not yet fully moved through the model. The apparent increase in variability of the $\lambda_{dem}$ in the 2010s could simply be related to additional variability in the estimated size of recent cohorts.

Using historical datasets intrinsically constrained the population growth rates estimated from the life-table analyses. For example, when the stock was at its highest abundance in the observed time series, $\lambda$, by definition, had to be $\leq 1$. Conversely, at the lowest abundance observed $\lambda$ had to be $\geq 1$. There were also more subtle constraints on the $\lambda$ estimates. For example, $\lambda$ had to be $\leq 2$ when a stock was at 50% of the maximum abundance, and the minimum value of $\lambda$ was similarly constrained by the minimum observed abundance observed in the time series. These constraints all contributed to an increase in the variability of $\lambda$ as abundance declined. However, they affected all stocks in the same manner and thus would not introduce systematic bias into our understanding of regional or species-specific patterns. In addition, we assumed the fishery removals in the stock assessments were correct and modified the productivity parameters to match the abundance time series from those stock assessment. This resulted in $\lambda_{dem}$ values that were 6.7% lower than those calculated using the initial productivity parameters found in the stock assessments. If this decline in $\lambda$ was instead attributed to fishing mortality (i.e., unobserved removals), then recovery potential in the absence of fishing would have increased (e.g., the doubling time would have declined).

Our life-table reconstruction enabled the underlying demographic rates to be estimated for stocks with sufficient age-structured data and provides an opportunity to compare the dynamics of these stocks using concepts developed in conservation biology and ecology. In doing so, our optimization procedure assumed that both the abundance estimates and the fishery removals from the stock assessment were correct. Although these assumptions are unlikely to be fully met, harvest advice is being derived for these stocks under the same general assumptions. Thus, our analysis reveals that current assessments can incorporate vastly different understandings of productivity and the potential for population growth for the same species across different regions, even when incorporating similar elements to describe population dynamics (e.g., equilibrium theory). Stocks characterized by low variability in population growth rates do not appear to be overly influenced by variation in environmental conditions (or that there was little environmental variability in the ecosystem), while those with a high innate capacity for population increase could indicate the opposite. However, it is not immediately apparent which conditions (e.g., environmental, ecological and/or anthropogenic) might be the primary determinants of population growth rates for a given stock, species, or system. Our findings underscore the need for alternative frameworks for deriving advice, particularly those that incorporate consideration of inter-species relationships and/or environmental conditions under the umbrella of ecosystem-based fishery management [82,83]. They also underscore the need to collect stock-specific data on marine fishes, rather than relying on species-level information from global repositories (e.g., Fishbase, Fishlife; [84]) or assuming values from closely related species to derive harvest advice (e.g., in cases in which there is insufficient data to estimate stock productivity, 79,[85]). While our analysis suggests that management advice may be fairly accurate for stocks with moderate variability in population growth rates or those that retain high innate capacity for population increase, it is likely to be the least accurate when recovery potential is low.

## Supporting information

**S1 Fig. Density dependence of $\lambda_{dem}$ for each stock in the analysis plotted regionally.**
(TIFF)

**S2 Fig. Density dependence of $\lambda_{real}$ for each stock in the analysis plotted regionally.**
(TIFF)

**S3 Fig. $R_0$ Estimates for the remainder of the stocks in this analysis (x-axis), plotted by the location of the stock(panels)** . The abbreviated species name is provided on the y-axis.
(TIFF)

**S4 Fig. $R_0$ Estimates for every cohort for each species excluding the effects of fishing.**
(TIFF)

**S5 Fig. $R_0$ Estimates for every cohort for each species including the effects of fishing.**
(TIFF)

**S6 Fig. Generation length estimates for the remainder of the stocks in this analysis (x-axis), plotted by the location of the stock (panels).** The abbreviated species name is provided on the y-axis.
(TIFF)

**S7 Fig. Generation length estimates for every cohort for each species excluding the effects of fishing.**
(TIFF)

**S8 Fig. Generation length estimates for every cohort for each species including the effects of fishing.**
(TIFF)

**S9 Fig. The number of stocks (-axis) which doubled in size after a given number of simulated years (x-axis) with no fishing mortality The dashed line represents the 77 stocks in the analysis.**
(TIFF)

**S10 Fig. Boxplots of $\lambda_{dem}$ estimates from the analysis in which the abundances were back-calculated for stocks in which the youngest age-class data available were greater than 1.** A subest of the results are shown here, with data for each stock of G. morhua, M. aeglefinus, P. virens, C. harengus, P. platessa, and S. solea. The blue stars indicate the mean and the dashed horizontal line represents a $\lambda$ of one. The y axis is on the log scale.
(TIFF)

**S11 Fig. Boxplots of $\lambda_{real}$ estimates from the analysis in which the abundances were back-calculated for stocks in which the youngest age-class data available were greater than 1.** A subest of the results are shown here, with data for each stock of G. morhua, M. aeglefinus, P. virens, C. harengus, P. platessa, and S. solea. The blue stars indicate the mean and the dashed horizontal line represents a $\lambda$ of one. The y axis is on the log scale.
(TIFF)

## Acknowledgments

This work would not have been possible without decades of research and data collection by untold numbers of people leading and supporting scientific surveys, data stewardship, methodological development, and research programs. We are in debt to everyone involved in the collection and analyses of these data. Funding for this project included support from the Canadian Scientific Research Funding program at Fisheries and Oceans Canada. In addition, this research was supported by the MAESTRO group funded by the synthesis center CESAB of the French Foundation for Research on Biodiversity (FRB; www.fondationbiodiversite.fr). We thank France Filière Pêche (FFP) who found the MAESTRO project.

## Author contributions

**Conceptualization:** David M. Keith, Heather D. Bowlby, Arnaud Auber.

**Data curation:** David M. Keith, Julie A Charbonneau.

**Formal analysis:** David M. Keith, Heather D. Bowlby.

**Investigation:** David M. Keith.

**Methodology:** David M. Keith, Heather D. Bowlby, Danielle M Baribeau.

**Project administration:** David M. Keith.

**Software:** David M. Keith.

**Supervision:** David M. Keith.

**Validation:** David M. Keith.

**Visualization:** David M. Keith, Danielle M Baribeau, Julie A Charbonneau, Freya Keyser.

**Writing – original draft:** David M. Keith, Heather D. Bowlby, Julie A Charbonneau, Freya Keyser.

**Writing – review & editing:** David M. Keith, Heather D. Bowlby, Camille Albouy, Arnaud Auber, Danielle M Baribeau, Daniel G Boyce, Julie A Charbonneau, Freya Keyser, Kristin M Kleisner, Martin P Marzloff, Katherine E Mills, David Mouillot, Aurore Receveur, Nancy L Shackell, Rita P Vasconcelos.

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
