## [Decision Letter · Decision Letter 0]

14 Nov 2025

Dear Dr. Keith,

Thank you for submitting your manuscript to PLOS ONE. After careful consideration, we feel that it has merit but does not fully meet PLOS ONE’s publication criteria as it currently stands. Therefore, we invite you to submit a revised version of the manuscript that addresses the points raised during the review process. Both reviewers suggested minor changes and made detailed comments that should be carefully considered. However, the comments of one of the reviewers imply moderate revision, which means that the revised manuscript will be sent back to the reviewers for a second round of revision.

We look forward to receiving your revised manuscript.

Kind regards,

Athanassios C. Tsikliras

Academic Editor

PLOS ONE

**Journal Requirements:**

2. Please update your submission to use the PLOS LaTeX template. The template and more information on our requirements for LaTeX submissions can be found at http://journals.plos.org/plosone/s/latex .

3. We note that Figure 1 in your submission contain map images which may be copyrighted. All PLOS content is published under the Creative Commons Attribution License (CC BY 4.0), which means that the manuscript, images, and Supporting Information files will be freely available online, and any third party is permitted to access, download, copy, distribute, and use these materials in any way, even commercially, with proper attribution. For these reasons, we cannot publish previously copyrighted maps or satellite images created using proprietary data, such as Google software (Google Maps, Street View, and Earth). For more information, see our copyright guidelines: http://journals.plos.org/plosone/s/licenses-and-copyright.

A. You may seek permission from the original copyright holder of Figure(s) [#] to publish the content specifically under the CC BY 4.0 license. 

B. If you are unable to obtain permission from the original copyright holder to publish these figures under the CC BY 4.0 license or if the copyright holder’s requirements are incompatible with the CC BY 4.0 license, please either i) remove the figure or ii) supply a replacement figure that complies with the CC BY 4.0 license. Please check copyright information on all replacement figures and update the figure caption with source information. If applicable, please specify in the figure caption text when a figure is similar but not identical to the original image and is therefore for illustrative purposes only.

4. Please remove your figures from within your manuscript file, leaving only the individual TIFF/EPS image files, uploaded separately. These will be automatically included in the reviewers’ PDF.

5. Please include captions for your Supporting Information files at the end of your manuscript, and update any in-text citations to match accordingly. Please see our Supporting Information guidelines for more information: http://journals.plos.org/plosone/s/supporting-information .

Reviewers' comments:

Reviewer's Responses to Questions

**Comments to the Author**

1. Is the manuscript technically sound, and do the data support the conclusions?

Reviewer #1: Yes

Reviewer #2: Partly

2. Has the statistical analysis been performed appropriately and rigorously?

Reviewer #1: Yes

Reviewer #2: Yes

3. Have the authors made all data underlying the findings in their manuscript fully available?

Reviewer #1: No

Reviewer #2: Yes

4. Is the manuscript presented in an intelligible fashion and written in standard English?

Reviewer #1: Yes

Reviewer #2: Yes

Reviewer #1: Demographics and recovery potential of globally exploited marine teleosts by Keith et al.

In this manuscript, the authors aim to carry out a model-based study to gain insight into the roles of life history traits and density dependence in the population growth, recovery potential, and demographic variability of commercially harvested fish stocks in the North Atlantic and North Pacific. Specifically, this study evaluates population and demographic responses to fishing pressure for 77 marine fish stocks in the regions by reconstructing life tables using Leslie matrices with annual demographic parameters (fecundity and survivorship) estimated from age-structured stock assessment model outputs. The authors conclude that most of the stocks in the study (nearly 90%) have a good recovery potential, but both population growth and recover potential can be highly variable among stocks, indicating weak evidence of life history characteristics’ roles in population and demographic variability.

Overall, this is a very well-written manuscript with clearly stated objectives that are worthy of investigation. I believe that the authors present interesting, solid work that provides much needed information in evaluating the recovery potential of harvested fish stocks exposed to varying levels of fishing pressure. While the roles of demographic variability and density dependence in the population dynamics of exploited fish species have been explored in past research, by applying the life table approach to a large number of stocks in several regions this study provides further evidence that the commonly held view of life history theory may need to be reconsidered when applied to harvested species. I just have a few minor comments below.

Specific comments:

l.19: ‘stocks’ -> ‘stocks with’

l.98: Has this repository become publicly available?

l.126-l.129: I would suggest the authors add the results of this test to the Supplementary material.

l.231: Was density dependence tested on lambda in the same year? I would imagine there is a time lag of a year (or more) at least in some stocks.

l.251: Does the interaction here mean ‘Stock’ and ‘Prop’?

l.347: ‘it’s’ -> ‘its’

l.406: ‘stock’ -> ‘stocks’.

l.518: ‘comnpare’ -> ‘compare’

Reviewer #2: The authors apply the Lotka–Euler equation to life-history schedules derived from stock assessment outputs to estimate demographic growth capacity for 77 stocks in the North Atlantic and Northeast Pacific, both including and excluding the effects of fishing mortality. They also test whether the populations show signs of density dependent growth rate and they also estimated various demographic parameters, like lifetime reproductive success, generation and doubling time. Based on these estimates, they comment on the innate capacity of these stocks to recover as well as on the drivers of variation in population productivity under different levels of exploitation and density conditions. Overall, the manuscript addresses an important question: how well demographic life-table metrics deriving from standard stock assessment outputs reflect the potential for growth and recovery in exploited marine fish populations. The idea is timely, the analyses are methodologically sound, and the authors are careful in presenting caveats associated with their approach.

In any case, the weaknesses of this analysis are explicitly pointed out by the authors themselves. Probably the most striking outcome is the almost complete lack of intra- and interspecific patterns consistent with life-history theory expectations. For example, slow growing, high trophic level species like cod or haddock showed comparable or even higher growth capacity and recovery potential than fast growing, lower trophic level species such as herring or sardine. This has led the authors to acknowledge that probably “equilibrium theory and general life-history characteristics […] are poor predictors of population growth rates and recovery potential.”

However, probably the main reason for these inconsistencies does not lie in the authors’ analytical approach, but in the nature of the input data. Stock assessments are designed primarily to provide management-relevant estimates of biomass, fishing mortality, and abundance trends. While model diagnostics focus on statistical fit, assessment models often rely on biological assumptions that are not always inspected or validated, potentially resulting in biologically unrealistic representations of population dynamics. Moreover, outputs such as abundance-at-age, maturity, natural mortality, and survival are influenced by model structure, data limitations, retrospective adjustments, and exploitation history. These limitations are acknowledged by the authors in several parts of the manuscript, particularly in the final paragraph of the Discussion.

Nonetheless, these limitations do not diminish the value of the manuscript. The underlying idea is solid and interesting, the analyses are robust, and the text is well written and clearly justified. Importantly, the authors show that under current management regimes stocks still rely on recruitment bonanzas for retaining production capacity, a critical message for designing sustainable management strategies. Their density-dependence analysis is also interesting and in line with ecological and demographic theory.

Thus, I recommend publication of this paper after revision.

Specific Comments

Title

The authors analyze 77 stocks from the North Atlantic and Northeast Pacific Oceans. Given this regional coverage, the phrase “globally exploited marine teleosts” overstates the spatial extent and representativeness of the dataset. I recommend rephrasing the title to better reflect the actual geographical scope of the analysis.

Lines 83 – 102:

The dataset used by the authors (Charbonneau and Keith 2022), available through the GitHub repository cited in the corresponding reference, contains information on abundance, weight-at-age, fishing mortality, and Fmsy (for some stocks) for 101 stocks. This dataset does not seem to include information on maturity, natural mortality and fishery removals, so it would be better if the authors explained the source for this data too. In addition, since that dataset includes more stocks than the 77 used here, please explain the criteria or data filtering steps that led to your final selection.

Line 112:

Since you apply the discrete time version of the Euler-Lotka equation, I don’t think that you need dx in your equation.

Lines 160 – 187:

The authors refer to fishing mortality, natural mortality and total mortality, but the values used in the Euler–Lotka equations are actually annual proportional survival or removal rates (bounded between 0 and 1), rather than instantaneous mortality rates (F, M, Z) in the standard continuous-time fisheries sense/stock assessment routine.

So, it would be important for the authors to explicitly state that mortality inputs were transformed into proportional form before being used in the life-table calculations to help avoid possible confusion, especially for readers accustomed to interpreting F, M, and Z as instantaneous rates in stock assessment models.

Lines 360 – 361:

A verb is missing from the sentence “The changes in the rate of…”

Lines 435 – 436:

The authors state that “when the effects of fishing were excluded, these results indicated these stocks retained an innate capacity for increase of approximately 8.7%”. This value is derived from the median population growth rate when excluding the effects of fishing for the 77 stocks (lines 321-322). However, λdem, as correctly pointed out by the authors, is not independent from fishing pressure and by no means represent the maximum intrinsic rate of growth for populations.

By relying on the median λdem over the full time series, the authors implicitly assume that fishing pressure, status and management of stocks have been relatively stable throughout the nearly 40 years analyzed. However, no information is provided on whether there are temporal trends for λ stock values. If there is an increasing or decreasing trend for λ (which seems plausible based on changing exploitation levels or even environmental conditions), then the median across all years may not reflect the current innate capacity for population change.

In that case, probably the current (or recent) capacity of the populations cannot support an 8.7% increase. If such temporal trends actually exist, then a metric based on more recent years (e.g., the mean or median of the last three to five years) may offer a more accurate estimate of present-day demographic potential. I suggest authors to examine and report whether temporal trends exist and possibly revise their assumptions accordingly.

Line 518:

Correct the typo “comnpare”

Tables 1 and 2:

I assume that the authors chose to estimate and present λ at 40% of maximum abundance because this is the threshold where the density independent part of the model begins. However, this is not stated explicitly anywhere in the manuscript, nor in the tables captions. I suggest to add a note (within the text or/and in table caption) explaining the choice of 40% estimations.

Figures:

Please italicize the species names.

**Do you want your identity to be public for this peer review?** For information about this choice, including consent withdrawal, please see our Privacy Policy

Reviewer #1: No

Reviewer #2: No

---

## [Decision Letter · Decision Letter 1]

21 Dec 2025

Demographics and recovery potential of exploited marine teleosts

PONE-D-25-42451R1

Dear Dr. Keith,

We’re pleased to inform you that your manuscript has been judged scientifically suitable for publication and will be formally accepted for publication once it meets all outstanding technical requirements.

Kind regards,

Athanassios C. Tsikliras

Academic Editor

PLOS One

Additional Editor Comments (optional):

Reviewers' comments:

Reviewer's Responses to Questions

**Comments to the Author**

Reviewer #2: All comments have been addressed

2. Is the manuscript technically sound, and do the data support the conclusions?

Reviewer #2: Yes

3. Has the statistical analysis been performed appropriately and rigorously?

Reviewer #2: Yes

4. Have the authors made all data underlying the findings in their manuscript fully available?

Reviewer #2: Yes

5. Is the manuscript presented in an intelligible fashion and written in standard English?

Reviewer #2: Yes

Reviewer #2: The authors have done an excellent job addressing my comments. I recommend that the manuscript be accepted for publication.

**Do you want your identity to be public for this peer review?** For information about this choice, including consent withdrawal, please see our Privacy Policy

Reviewer #2: No

---

## [Editor Report · Acceptance letter]

PONE-D-25-42451R1

PLOS One

Dear Dr. Keith,

I'm pleased to inform you that your manuscript has been deemed suitable for publication in PLOS One. Congratulations! Your manuscript is now being handed over to our production team.

Kind regards,

on behalf of

Professor Athanassios C. Tsikliras

Academic Editor

PLOS One